# Modeling organic aerosol over Europe in summer conditions with the VBS-GECKO parameterization: sensitivity to secondary organic compound properties and IVOC emissions

Victor Lannuque[1,2,3,a], Florian Couvidat[2], Marie Camredon[1], Bernard Aumont[1] and Bertrand Bessagnet[2,b]

[1] LISA, UMR CNRS 7583, IPSL, Université Paris Est Créteil and Université de Paris, 94010 Créteil Cedex, France.
[2] INERIS, National Institute for Industrial Environment and Risks, Parc Technologique ALATA, 60550 Verneuil-en-Halatte, France.
[3] Agence de l'Environnement et de la Maîtrise de l'Energie, 20 avenue du Grésillé - BP 90406, 49004 Angers Cedex 01, France.
[a] Now at: CEREA, Joint Laboratory École des Ponts ParisTech – EDF R&D, Université Paris-Est, 77455 Marne la Vallée, France.
[b] Now at: Laboratoire de Métérologie Dynamique, IPSL, CNRS, UMR8539, 91128 Palaiseau Cedex, France.

*Correspondence to*: Victor Lannuque (victor.lannuque@lisa.u-pec.fr) and Florian Couvidat (florian.couvidat@ineris.fr)

**Abstract.** The VBS-GECKO parameterization for secondary organic aerosol (SOA) formation was integrated in the chemistry-transport model CHIMERE. Concentrations of organic aerosol (OA) and SOA were simulated over Europe for the July-August 2013 period. Simulated concentrations with the VBS-GECKO were compared to results obtained with the former $H^2O$ parameterization implemented in CHIMERE and to observations from EMEP, ACTRIS and other observations available in the EBAS database. The model configuration using the VBS-GECKO parameterization slightly improves the performances compared to the model configuration using the former $H^2O$ parameterization. The VBS-GECKO model configuration performs well for stations showing a large SOA concentration from biogenic sources, especially in northern Europe, but underestimates OA concentrations over stations close to urban areas. Simulated OA was found to be mainly secondary (~85%) and from terpene oxidation. Simulations show negligible contribution of the oxidation of mono-aromatic compounds to SOA production. Tests performed to examine the sensitivity of simulated OA concentrations to hydro-solubility, volatility, aging rates and $NO_x$ regime have shown that the VBS-GECKO parameterization provides consistent results, with a weak sensitivity to changes in the parameters provided by the gas phase mechanism included in CHIMERE (e.g. $HO_x$ or $NO_x$ concentrations). Different scenarios considering intermediate-volatility organic compound (IVOC) emissions were tested to examine the contribution of IVOC oxidation to SOA production. At the continental scale, these simulations show a weak sensitivity of OA concentrations to IVOC emission variations. At the local scale, accounting for IVOC emissions was found to lead to a substantial increase of OA concentrations in the plume from urban areas. This additional OA source remains too small to explain the gap between simulated and measured values at stations where anthropogenic sources are dominant.

## 1. Introduction

For the past 20 years, fine particulate matter or $PM_{2.5}$ (particles with a diameter smaller than 2.5 µm) has been regulated due to their health impacts and the resulting costs (e.g. Lim et al., 2012; WHO Regional Office for Europe and OECD, 2015). Furthermore, fine particles degrade visibility (e.g. Han et al., 2012) and influence climate change (e.g. Boucher et al., 2013). Organic aerosol (OA) represents a large fraction of the total fine particle mass (e.g. Jimenez et al., 2009). This OA is either primary (directly emitted into the atmosphere) or secondary (formed by gas/particle partitioning of low volatile and/or highly soluble species produced during the oxidation of gaseous organic compounds) (e.g. Carlton et al., 2009; Kroll and Seinfeld, 2008). The secondary organic aerosol (SOA) dominates the primary organic aerosol (POA) in most environments (e.g. Gelencsér et al., 2007; Jimenez et al., 2009).

Chemistry-transport models (CTMs) are used to investigate and identify air quality regulation policies. Parameterizations are developed and used in CTMs to represent SOA formation. Different approaches have been followed to describe SOA formation as the two-product model (e.g. Odum et al., 1996; Schell et al., 2001), the molecular approach (e.g. Pun et al., 2002, 2003), the volatility basis set (VBS) approach (e.g. Donahue et al., 2006, 2012) or the statistical oxidation model (SOM) (e.g. Cappa and Wilson, 2012; Jathar et al., 2015). Parameterizations are constantly improved and additional processes were included in the parameterizations to improve the simulations of SOA concentrations, such as gas-phase aging of organic species (e.g. Rudich et al., 2007), more comprehensive emissions and multiphase chemistry. Robinson et al. (2007) have indeed shown that POA provided in emission inventories is in part composed of semi-volatile organic compounds (SVOCs) (existing both in particle and gas phases) and that a fraction of emitted organic compounds were missing from these inventories: the intermediate-volatility organic compounds (IVOCs) (forming SOA after several oxidation stages) (e.g. Ots et al., 2016; Robinson et al., 2007; Woody et al., 2015). Numerous experimental and modeling studies have since explored the volatility distribution of SVOCs from POA emissions and of IVOC emissions depending on the emission source (e.g. Akherati et al., 2019; Grieshop et al., 2009; Hatch et al., 2018; Jathar et al., 2017; Louvaris et al., 2017; Lu et al., 2018; May et al., 2013a, 2013b, 2013c; Woody et al., 2016).

Other studies have highlighted the important role played by condensed phase processes in SOA formation, in particular the reactivity of hydrophilic products in the condensed phase (e.g. Couvidat et al., 2012; Couvidat and Seigneur, 2011; Knote et al., 2014; Paulot et al., 2009; Pun et al., 2006b; Surratt et al., 2010), the oligomerization of SVOCs in the aerosol (e.g. Aksoyoglu et al., 2011; Couvidat et al., 2012; Denkenberger et al., 2007; Dommen et al., 2006; Kalberer et al., 2006; Lemaire et al., 2016; Trump and Donahue, 2014), the non-ideal behavior of the organic aerosol (Couvidat et al., 2012, Couvidat and Sartelet, 2015; Pun et al., 2006; Pye et al., 2018) or the effect of the aerosol viscosity (Couvidat and Sartelet, 2015; Shiraiwa et al., 2013). Comparisons with field observations have shown that CTMs using these parameterizations fall short to reproduce SOA concentration spatial and temporal variability (e.g. Aksoyoglu et al., 2011; Bessagnet et al., 2016;

Ciarelli et al., 2016; Couvidat et al., 2012; Heald et al., 2005; Im et al., 2015; Petetin et al., 2014; Pun et al., 2006a; Solazzo et al., 2012; Tsigaridis et al., 2014; Volkamer et al., 2006).

Most of these SOA parameterizations are optimized/built on the basis of atmospheric chamber data. Experiments are however limited in number and are usually performed under conditions that differ from the atmosphere. In addition, SOA formation experiments can be subject to potential artefacts from chamber wall surfaces, such as aerosol and gaseous compound wall losses (e.g. La et al., 2016; Matsunaga and Ziemann, 2010; McMurry and Grosjean, 1985). Considering or not these artefacts for the parameterization development directly impact SOA representation in air quality models (e.g. Cappa et al. 2016).

The development of the VBS-GECKO parameterization explores another track using the results of an explicit model representing the state of knowledge of organic gas-phase chemistry instead of atmospheric chamber data. The VBS-GECKO parameterization for SOA formation (Lannuque et al., 2018) is a VBS-type parameterization with gaseous aging. VBS-GECKO was optimized based on box modeling results using explicit oxidation mechanisms generated with the Generator for Explicit Chemistry and Kinetics of Organics in the Atmosphere (GECKO-A) modeling tool (Aumont et al, 2005; Camredon et al, 2007).

The objectives of this study are (i) to evaluate the behavior of the VBS-GECKO parameterization in the CTM CHIMERE (Menut et al., 2013, Mailler et al., 2017) by comparison with field measurements and previous simulations obtained with the $H^2O$ parameterization (Couvidat et al., 2012) already implemented in CHIMERE, (ii) to explore the sensitivity of simulated SOA concentrations to organic compound properties (volatility, solubility, aging rates or $NO_x$ regime) and (iii) to test the sensitivity of OA concentrations to the uncertainties on IVOC emission fluxes from traffic,. The setup of the CHIMERE model and the implementation of the VBS-GECKO in the CTM are described in section 2. In section 3, the VBS-GECKO is evaluated over Europe for a two-month summer period, the sensitivity to organic compound properties is explored in section 4, and the sensitivity to the uncertainties in IVOC emission fluxes from traffic is investigated in section 5. Finally, results on simulated OA sources and concentrations are discussed in section 6.

## 2. Method

### 2.1. The CHIMERE chemical transport model

The evaluation of the VBS-GECKO parameterization and the exploration of SOA sensitivity were performed using the CHIMERE 2017 β version. This version is based on the CHIMERE 2013 version (Menut et al., 2013) which was modified to improve the representation of particles with the implementation of a new aerosol module. Details of the CHIMERE 2017 β version and its evaluation are given in Couvidat et al. (2018).

Briefly, the CHIMERE 2017 β version uses the MELCHIOR2 gas-phase chemical scheme, involving 44 species reacting according to 120 reactions. MELCHIOR2 is a reduced version of the MELCHIOR1 mechanism, obtained by the Carter's surrogate molecule method (Carter, 1990). In CHIMERE, the aerosol evolution is described by a sectional aerosol module

(e.g. Bessagnet et al., 2004, 2009; Schmidt et al., 2001). The size distribution of aerosol particles is here represented using 9 bins, ranging from 10 nm to 10 µm. Aerosol formation is represented in the model by nucleation for sulfuric acid (Kulmala et al., 1998), coagulation between particles (e.g. Debry et al., 2007; Jacobson et al., 1994) and condensation/evaporation via absorption according to the "bulk equilibrium" approach (e.g. Pandis et al., 1993). For inorganic species, the gas/particle

equilibrium concentrations are calculated using the ISORROPIA v2.1 (Fountoukis and Nenes, 2007) thermodynamic module. For organic species, the equilibrium concentrations are calculated using the SOAP (Secondary Organic Aerosol Processor) thermodynamic module (Couvidat and Sartelet, 2015). The gaseous formation of secondary organic species able to partition between the gas and the condensed phases (so leading to SOA formation) are represented in the CHIMERE β version using the $H^2O$ mechanism. Here, the VBS-GECKO parameterization was also implemented. The $H^2O$ and VBS-

GECKO organic aerosol modules are described hereafter.

The chemical speciation of emitted non-methane volatile organic compounds (NMVOCs) is taken from Passant (2002) as described in Menut et al. (2013). POA from emission inventories are considered as SVOC. A factor 5 is apply to residential POA emissions, as wood burning emissions are underestimated in emission inventories (e.g. Denier Van Der Gon et al., 2015). This factor was shown to give satisfactory results on OA estimations (Couvidat et al. 2012, 2018). Biogenic emissions

are computed with the Model of Emissions and Gases and Aerosols from Nature MEGAN 2.1 algorithm (Guenther et al., 2012). Dry deposition for gaseous organic species is described using the Wesely (1989) parameterization and according to their Henry's law constants, as described by Bessagnet et al. (2010).

## 2.2. The organic aerosol modules

The purpose of the comparison between the $H^2O$ and GECKO-VBS mechanisms for SOA formation is to evaluate the

reliability of the VBS-GECKO parameterization. The simulations performed with CHIMERE were therefore setup using the same configuration of the model (meteorological data, emissions, deposition, inorganic and organic gaseous chemical mechanism, inorganic and organic gas/particle partitioning…), but implementing either the $H^2O$ or the GECKO-VBS parameterizations. The implementation of a given parameterization for SOA formation induces anyway some differences, related to the primary compounds considered and/or the processes taken into account in the parameterization. The

differences between the $H^2O$ and VBS-GECKO are mentioned in the following parameterization presentation sections.

### 2.2.1. The $H^2O$ reference mechanism

The $H^2O$ mechanism, described in details by Couvidat et al. (2018), considers SOA formation from the partitioning of hydrophilic species (condensing on an aqueous phase and an organic particulate phase) and hydrophobic species (condensing only on an organic particulate phase owing to their low affinity with water). Distinction between hydrophobic and

hydrophilic compounds is based on their octanol/water coefficient (Pun et al., 2006) or their partitioning between the organic and aqueous phases (Couvidat and Seigneur, 2011). $H^2O$ considers the formation of hydrophilic and/or hydrophobic species from the gaseous oxidation of isoprene, monoterpenes (α-pinene, β-pinene, limonene and ocimene), sesquiterpenes

(humulene) and mono-aromatic precursors (toluene and xylenes). Note that in H$^2$O, limonene mechanism is used as a surrogate mechanism for ocimene (ocimene having its own OH, NO$_3$ and O$_3$ reaction rates). Each emitted SOA precursor is linked to a species of the H$^2$O mechanism. POA provided by emissions inventories are split into three emitted SVOCs having different volatilities (saturation vapor pressures at 298 K of 8.9×10$^{-11}$, 8.4×10$^{-9}$ and 3.2×10$^{-7}$ atm respectively) with a fraction

that follows the volatility distribution of POA emissions given by Robinson et al. (2007). In H$^2$O, gaseous oxidation of these three compounds with OH leads to hydrophobic species with a lower volatility. No gaseous oxidation is considered for hydrophylic and hydrophobic species in H$^2$O. Activity coefficients for the H$^2$O species are computed with the thermodynamic model UNIFAC (UNIversal Functional group Activity Coefficient; Fredenslund et al., 1975). H$^2$O has been evaluated over Europe (Couvidat et al., 2012, 2018) and the Paris area (Couvidat et al., 2013; Zhu et al., 2016a, 2016b). The

H$^2$O reference mechanism is presented in table S1 of supplementary material.

### 2.2.2.  The VBS-GECKO parameterization

The VBS-GECKO parameterization is described in details in a previous paper by Lannuque et al. (2018). Briefly, VBS-GECKO is a volatility basis set (VBS) type parameterization that represents SOA formation from the partitioning of organic compounds having a low volatility onto an organic aerosol phase. The VBS-GECKO parameterization takes into account for

the oxidation of a precursor k (precu$_k$) (1) the formation of 7 VB$_{k,i}$, where i is the number of the volatility bin (1 being the most volatile and 7 the less volatile) (reactions R1, R2 and R3), (2) the gas-phase ageing of the VB$_{k,i}$ (except for the lowest volatility bin 7) with OH redistributing the matter between the VB$_{k,i}$ (reactions R4) and by photolysis leading to a loss of carbon matter (reactions R4), (3) the gas/particle partitioning of the precursor k (R6) and of the VB$_{k,i}$ (R7) ; the VBS-GECKO follows this structure for a given precursor k:

$$\mathbf{precu_k^{(g)} + OH \rightarrow a_{k,RRR,1} VB_{k,1} + a_{k,RRR,2} VB_{k,2} + ... + a_{k,RRR,n} VB_{k,7}} \qquad \mathbf{k_{precuk+OH}} \qquad (R1)$$

$$\mathbf{precu_k^{(g)} + O_3 \rightarrow b_{k,RRR,1} VB_{k,1} + b_{k,RRR,2} VB_{k,2} + ... + b_{k,RRR,n} VB_{k,7}} \qquad \mathbf{k_{precuk+O3}} \qquad (R2)$$

$$\mathbf{precu_k^{(g)} + NO_3 \rightarrow c_{k,RRR,1} VB_{k,1} + c_{k,RRR,2} VB_{k,2} + ... + c_{k,RRR,n} VB_{k,7}} \qquad \mathbf{k_{precuk+NO3}} \qquad (R3)$$

$$\mathbf{VB_{k,i}^{(g)} + OH \rightarrow d_{k,RRR,i,1} VB_{k,1} + d_{k,RRR,i,2} VB_{k,2} + ... + d_{k,RRR,i,n} VB_{k,7}} \qquad \forall \mathbf{i \neq 7} \quad \mathbf{k_{OH}} = 4.10^{-11} \text{cm}^3 \text{molec}^{-1}\text{s}^{-1} \quad (R4)$$

$$\mathbf{VB_{k,i}^{(g)} + h\nu \rightarrow \ carbon\ lost} \qquad \forall \mathbf{i \neq 7} \quad \boldsymbol{\phi}_k \mathbf{J_{acetone}} \qquad (R5)$$

$$\mathbf{precu_k^{(g)} \leftrightarrow precu_k^{(p)}} \qquad (R6)$$

$$\mathbf{VB_{k,i}^{(g)} \leftrightarrow VB_{k,i}^{(p)}} \qquad (R7)$$

In VBS-GECKO, the production and gaseous aging of the VB$_{k,i}$ for a precursor k are adjusted by stoichiometric coefficients (a$_{k,RRR,i}$, b$_{k,RRR,i}$, c$_{k,RRR,i}$, d$_{k,RRR,i}$, for reaction (R1) to (R4) respectively) which depend on NO$_x$ regime. The formation of more volatile and less volatile bins can be assimilated to fragmentation and functionalization processes, respectively. The

stoichiometric coefficients depend on the NO$_x$ according to the reaction rate ratio (RRR) of RO$_2$ with NO:

$$\text{RRR} = \frac{k_{RO_2+NO}[NO]}{k_{RO_2+NO}[NO] + k_{RO_2+HO_2}[HO_2]} \qquad (1)$$

where $k_{RO2+NO}$ (set to $9.0 \times 10^{-12}$ cm$^3$ molec$^{-1}$ s$^{-1}$ according Jenkin et al., 1997 at 298 K) and $k_{RO2+HO2}$ (set to $2.2 \times 10^{-11}$ cm$^3$ molec$^{-1}$ s$^{-1}$ according to Boyd et al., 2003, assuming a large carbon skeleton for RO$_2$ at 298 K) are the rate constants for the reactions of the peroxy radicals with NO, and HO$_2$ respectively and [NO] and [HO$_2$] the concentration of the radicals. The entire RRR range is covered by linear interpolation of the coefficients between the two closest values. The photolysis is considered as a limiting process for SOA formation, leading to a loss of matter. The photolysis rate of the VB$_{k,i}$ are based on the acetone one multiplied by an optimized factor $\phi_k$, different for each precursor k. Precursors and VB$_{k,i}$ condense on an organic particulate phase according to an equilibrium between the gas and the organic particulate phase that follows the Raoult's law (reaction R6 and R7).

The properties of the 7 VB$_{k,i}$ were considered to be independent of the precursor k and set for each volatility bin i to the mean values simulated with explicit GECKO-A simulations. Table 1 gives the molar weights (Mw), saturation vapor pressures (P$^{sat}$) at 298 K, effective Henry's law constants (H$^{eff}$) at 298K and vaporizations enthalpies ($\Delta$H$_{vap}$) used for each VB$_{k,i}$ VBS-GECKO species. The stoichiometric coefficients and factors $\phi_k$ were optimized on explicit GECKO-A simulations of gas-phase oxidation and SOA formation. The stoichiometric coefficients were optimized for 5 RRR values: 0, 0.1, 0.5, 0.9 and 1 (Lannuque et al., 2018). Precursors considered in the current VBS-GECKO parameterization are mono-aromatic compounds (benzene, toluene, and o- m- and p-xylenes) and n-alkanes (decane, tetradecane, octadecane, docosane and hexacosane) reacting with OH, and monoterpenes ($\alpha$-pinene, $\beta$-pinene and limonene) and linear 1-alkenes (decene, tetradecene, octadecene, docosene and hexacosene) reacting with OH, O$_3$ and NO$_3$. Note that (1) the parameterization does not represent SOA formation from the partitioning of hydrophilic species, (2) recently indentified chemical processes such as autooxidation reactions or acid-catalyzed pathways, not include in GECKO-A, are thus not considered in the VBS-GECKO parameterization and (3) the high value of the reaction rate of the VB$_{k,i}$ ($k_{OH} = 4.10^{-11}$ cm$^3$ molec$^{-1}$ s$^{-1}$) was fixed before optimization and is compensated by lower or higher values of optimized coefficients (see details in Lannuque et al., 2018). Tables of optimized stoichiometric coefficients are available in supplementary material of Lannuque et al. (2018).

For SOA production from NMVOC oxidation, the former H$^2$O parameterization in CHIMERE was replaced by the VBS-GECKO parameterization for terpenes and mono-aromatic compounds. The VBS-GECKO mechanisms were also implemented in CHIMERE for SOA formation from C10 to C13 alkanes and alkenes, gaseous species usually not considered in 3D models as SOA precursors. Each emitted SOA precursor not present in the VBS-GECKO was linked to a VBS-GECKO species. As in the H$^2$O mechanism, the VBS-GECKO parameterization for limonene was used as a surrogate mechanism for ocimene. The VBS-GECKO parameterizations for benzene, toluene, o-, m- and p-xylenes were also used as surrogate mechanisms for other emitted mono-aromatic compounds according to their SOA yield and reactivity with OH. n-dodecane and tetradecane VBS-GECKO species were used to lump emitted alkanes with 10 to 13 atoms of carbon, according to their carbon chain length. The VBS-GECKO mechanism for 1-decene was applied for all emitted C10 alkenes. The lumping scheme between emitted NMVOCs and VBS-GECKO species is given in Table 2. The current VBS-GECKO version does not represent SOA production from the oxidation of isoprene and sesquiterpenes. The H$^2$O parameterizations for isoprene and humulene were therefore left unchanged in CHIMERE to account for this SOA production. For SOA

production for SVOC oxidation distributed from POA emissions, the $H^2O$ approach was kept unchanged (i.e. distribution of POA emissions into 3 SVOC species and representation of their SOA production using the $H_2O$ mechanism). In CHIMERE, RRR is calculated in each box at each chemical time step following equation 1. Activity coefficients for the condensation of the VBS-GECKO species into the aerosol particulate phase are fixed to 1 (i.e. ideality of the organic particulate phase is

considered). This implementation of the VBS-GECKO in CHIMERE was selected here as the reference configuration and is denoted ref-VBS-GECKO hereafter. The ref-VBS-GECKO mechanism is presented in table S2 of supplementary material and evaluated in section 3.

Changes were then applied to this reference configuration to perform sensitivity tests of SOA formation on secondary organic compound properties (solubility, reactivity with OH, $NO_x/HO_2$ condition dependency and volatility) or IVOC

emission fluxes from traffic. For a better readability, the details of these modifications are presented for each sensitivity test in section 4 (properties) and section 5 (IVOC emissions).

### 2.3. Simulation setup and field measurements

The model was run to simulate the concentrations of OA over Europe (from 25° W to 45° E in longitude and from 30° to 70° N in latitude) with a horizontal resolution of 0.25° × 0.25° during the July-August 2013 period, SOA formation being

expected to be important during summertime. Meteorology was obtained from Integrated Forecasting System (IFS) model of the European Centre for Medium-Range Weather Forecasts (ECMWF). This meteorology has been evaluated in Bessagnet et al. (2016) for the model intercomparison project EURODELTA-III. ECMWF-IFS in the EURODELTA-III project has been shown to be one of the most reliable models to represent meteorological conditions over Europe. Anthropogenic emissions of gases and particles were taken from the European Monitoring and Evaluation Programme (EMEP) inventory

(methodology described in Vestreng, 2003) and boundary conditions were generated from the Model for OZone And Related Tracers (Mozart v4.0 (Emmons et al., 2010)). Wildfire emissions were not considered.

The VBS-GECKO mechanism was evaluated by comparing the simulated results to the $H^2O$ mechanism and particulate phase measurements available in the EBAS database (http://ebas.nilu.no/). EBAS is a database hosting observation data of atmospheric chemical composition and physical properties in support of a number of national and international programs

ranging from monitoring activities to research projects. EBAS is developed and operated by the Norwegian Institute for Air Research (NILU). This database is populated for example by the EMEP measurements (Tørseth et al., 2012) or the Aerosols, Clouds and Trace gases Research Infrastructure (ACTRIS, http://www.actris.eu/) ones. 48 rural background stations provide measurements for fine particulate matter and were thus selected here for a statistical evaluation: 36 stations for $PM_{2.5}$, 13 for $OC_{PM2.5}$ (organic carbon in $PM_{2.5}$, obtained by filter calcinations) and 6 for $OM_{PM1}$ (organic matter in $PM_1$, obtained with

ACSM). For the comparisons with OC measurements, the OM:OC ratio of the VBS-GECKO volatility bins were assumed to be equal to 1.8, in agreement with typical observed values given by Canagaratna et al. (2015). The location of the selected stations is shown in Fig. 1.a. Among these stations, 7 stations were used for time series comparisons:

- the Cabauw (NL0644R, Netherlands), Melpitz (DE0044R, Germany) and Palaiseau (FR0020R, SIRTA, France) rural background stations, located in areas dominantly impacted by anthropogenic air masses (see Figure S1 in supplementary material presenting the mean of the simulated ratios between toluene and α-pinene emission fluxes for the studied period).

- the Birkenes II (NO0002R, Norway), Diabla Gora (PL0005R, Poland), Hyytiälä (FI0050R, Finland) and Iskrba (SI0008R, Slovenia) rural background stations, located in areas dominantly impacted by biogenic emissions (see Figure S1 in supplementary material).

These 7 stations were selected among the 48 background station because the measurements at the station provide (1) a direct information on the organic fraction of fine particles, i.e. $OM_{PM1}$ and $OC_{PM2.5}$ measurements, and (2) enough data over the studied period to perform time series comparisons .The location of the 7 selected stations is shown in Fig. 1.b.

Various statistical indicators were computed to evaluate the VBS-GECKO mechanism, including the Root Mean Square Error (RMSE), the correlation coefficient, the Mean Fractional Error (MFE) and the Mean Fractional Bias (MFB). MFB and MFE are calculated as:

$$MFB = \frac{1}{N}\sum_{i=1}^{N} \frac{\left(c_i^{mod} - c_i^{obs}\right)}{\left(\frac{c_i^{mod} + c_i^{obs}}{2}\right)}, \qquad (2)$$

$$MFE = \frac{1}{N}\sum_{i=1}^{N} \frac{\left|c_i^{mod} - c_i^{obs}\right|}{\left(\frac{c_i^{mod} + c_i^{obs}}{2}\right)}, \qquad (3)$$

where $c_i^{mod}$ and $c_i^{obs}$ are the simulated and observed concentrations of the studied component at the time i. and N the number of available in-situ measurement values. Boylan and Russell (2006) defined two criteria to evaluate the performances of a model. The model performance criteria (described as the level of accuracy that is considered to be acceptable for modeling applications) is reached when MFE ≤ 75% and |MFB| ≤ 50% whereas the performance goal (described as the level of accuracy that is considered to be close to the best values a model can be expected to achieve) is reached when MFE ≤ 50% and |MFB| ≤ 30%. These criteria are currently used to evaluate the reliability of the models (e.g. Ciarelli et al., 2017; Couvidat et al., 2018; Lecœur and Seigneur, 2013; Mircea et al., 2019).

## 3. Evaluation of the ref-VBS-GECKO parameterization

Figure 2.a shows the mean OA mass concentrations simulated with the ref-VBS-GECKO version for the July-August 2013 period. The simulated mean OA concentrations range from ~0 µg.m$^{-3}$ in remote oceanic areas, to ~12 µg m$^{-3}$ around the Adriatic Sea and in the northern Italy, and are coherent with the expected orders of magnitude and spatial distributions over Europe (Aksoyoglu et al., 2011; Crippa et al, 2014). Figure 2.b presents the relative difference between mean OA mass concentrations simulated with ref-VBS-GECKO and with H$^2$O. The ref-VBS-GECKO produces more OA than H$^2$O, with a mean OA mass concentration around 30% higher on average over Europe. The increase is particularly important over northern Europe, with maximum differences reaching around +60%.

Table 3 gathers the statistical results calculated on daily averaged concentrations for ref-VBS-GECKO at the 48 stations (RMSE, Pearson's r, MFB and MFE), as well as the difference of this statistical indicator between ref-VBS-GECKO and $H^2O$. Statistical indicators show a high spatiotemporal correlation between ref-VBS-GECKO and measurements for daily $OM_{PM1}$ and $OC_{PM2.5}$ with r > 0.5 (0.79 and 0.57 respectively). These r values are in the standard of what has been found in previous modeling studies for Europe (Bergström et al., 2012; Ciarelli et al., 2017) or USA (Ahmadov et al., 2012; Murphy et al., 2017). For daily averaged measurements of $PM_{2.5}$, the correlation is smaller (0.42). This lower correlation for $PM_{2.5}$ has already been highlighted in summertime during the EURODELTA-III inter-comparison campaign (Bessagnet et al., 2016). MFE and MFB satisfy the performance criteria of Boylan and Russel (2006) for all the measurements. However, daily averaged $PM_{2.5}$, and especially the organic fraction ($OM_{PM1}$ and $OC_{PM2.5}$), appear to be systematically underestimated by the ref-VBS-GECKO model.

Comparing ref-VBS-GECKO statistical results with $H^2O$ statistical results, the simulated daily averaged $PM_{2.5}$ concentrations over the 36 stations appear to be weakly sensitive to the SOA formation mechanism used in the model. Only a slight improvement due to an increase in simulated $PM_{2.5}$ concentrations of about 5.5% is observed with the ref-VBS-GECKO model configuration. Concerning simulated $OC_{PM2.5}$ over the 13 measurement stations, the ref-VBS-GECKO parameterization leads to an increase of the simulated concentration (+ 15.6 %), ultimately leading to a clear improvement of MFE, MFB and correlation. Nevertheless, using the ref-VBS-GECKO configuration instead of the $H^2O$ configuration increases RMSE (+5%), owing to a substantial overestimation of OA. The main differences between the two organic aerosol modules are reached for $OM_{PM1}$ with simulated ref-VBS-GECKO concentrations higher than $H^2O$ by 31.5%. As simulated $OM_{PM1}$ concentrations were highly underestimated using the former $H^2O$ configuration compared to observations (6 stations), the ref-VBS-GECKO configuration improves RMSE, MFB and MFE.

Figure 3 shows comparisons between the measured and the simulated daily averaged temporal evolutions of $OM_{PM1}$, $OC_{PM2.5}$ and/or $PM_{2.5}$ concentrations at the 7 selected stations. Figure 4 shows the measured and simulated mean diurnal profiles at the 4 stations providing $OM_{PM1}$. Simulations capture qualitatively the observed feature of the daily averaged time series for $PM_{2.5}$, $OC_{PM2.5}$ and $OM_{PM1}$, and the mean diurnal profiles for $OM_{PM1}$. At stations dominantly impacted by biogenic sources, OA concentrations simulated with ref-VBS-GECKO are higher than those simulated with $H^2O$, leading to a better agreement with measurements (see Fig. 3.d to h and Fig. 4.c and d). However, day/night variations of $OM_{PM1}$ seem to be overestimated. At stations influenced by anthropogenic air masses, OA concentrations are weakly influenced by the organic aerosol module (see Fig. 3.a to c and Fig. 4.a and b). OA concentrations simulated with ref-VBS-GECKO are substantially underestimated, differences exceeding -50% for $OM_{PM1}$ concentrations at the Palaiseau and Melpitz stations as well as for $OC_{PM2.5}$ concentrations at the Cabauw station.

## 4. Sensitivity to the parameterization properties

Sensitivity tests were performed to assess the VBS-GECKO parameterization, evaluate the consistency of the modeling results and examine some hypotheses that may explain the gaps between measurement and simulated values. Sensitivities to hydro-solubility, gaseous aging, $NO_x$ regimes and volatility were studied comparing results to the non-modified ref-VBS-GECKO version.

### 4.1. Sensitivity tests to hydro-solubility and $H^{eff}$

SOA formation from the gas/particle partitioning of hydro-soluble organic compounds into an aqueous phase is now well recognized (e.g. Bregonzio-Rozier et al., 2016; Carlton et al., 2009; Knote et al., 2014). The effective Henry's law constant ($H^{eff}$) is the key parameter which controls this hydrophilic partitioning. Hydro-soluble organic compounds can also be lost at the surface by dry deposition. In CHIMERE, and according to the deposition scheme of Wesely (1989), the stomatal resistance of organic compounds depends on $H^{eff}$. To analyze the sensitivity of the simulated OA to hydrophilic partitioning and values of $H^{eff}$, the following two simulations were run:

- **Hydro-VBS-GECKO**. In this model configuration, $VB_{k,i}$ can condense both on organic and aqueous phases of particles. Aqueous-phase partitioning is computed according to Henry's law, assuming the particle phase behave as an ideal well mixed homogeneous aqueous phase. Deposition of $VB_{k,i}$ was already taken into account in the reference model configuration and was kept unchanged.

- **Hydro-VBS-GECKO-high**. This model configuration is identical to the hydro-VBS-GECKO configuration above, except that the original $H^{eff}$ of each $VB_{k,i}$ are multiplied by 100. The new $H^{eff}$ values correspond to the upper values of the $H^{eff}$ distribution of secondary organic compounds contributing to a given volatility bin (see Lannuque et al., 2018).

The relative difference on the simulated mean OA concentrations between Hydro-VBS-GECKO (respectively Hydro-VBS-GECKO-high) and ref-VBS-GECKO is given Fig. 5.a (respectively Fig. 5.b) for the two-month period. Figure 5.a shows that considering aqueous phase partitioning of the VBS-GECKO species leads to variations on the simulated mean OA concentrations below ±0.5%. Table 4 shows no significant modification in the statistical results for this simulation. The values of $H^{eff}$ set to each volatility bin increase when the volatility decreases (see Table 1), meaning that the less volatile species are also more prone to condense into the aqueous phase. Adding a hydrophilic partitioning does therefore not increase substantially the concentrations of organic species in the condensed phases.

The Hydro-VBS-GECKO-high configuration increases the mean simulated OA concentrations by ~10%, with a maximal increase reached over Belgium-Netherlands-Luxembourg area (called Benelux hereafter, around +20%, see Fig. 5.b). The contribution of the deposition and the partitioning processes are shown in Fig. 5.c and 5.d respectively. Changes due to deposition appear negligible (below ±0.2%) compared to the changes due to the aqueous partitioning (~+10%). According to the Wesely (1989) parameterization used for deposition, water solubility contributes to the surface resistance only. Knote et

al. (2014) have shown that deposition is not limited by the surface resistance for $H^{eff}$ greater than $10^8$ mol $L^{-1}$ atm$^{-1}$. In the ref-VBS-GECKO, this threshold corresponds to the $VB_{k,(3-7)}$ nominal $H_{eff}$ values, i.e. to the volatility bins partitioning mainly to OA. OA concentrations are therefore not sensitive to an increase of the $H_{eff}$ values. The increase of $H^{eff}$ by a factor of 100 makes possible hydrophilic partitioning of the most volatile bins that would not have condensed otherwise and leads to an increase of simulated OA concentrations. The maximum relative changes simulated over Benelux are mainly linked to the high relative humidity encountered in this area and the low simulated OA concentrations (see Fig. 2.a). This model configuration improves slightly the RMSE, MFB and MFE calculated on $OM_{PM1}$ (-4.58 %, -0.06 and -0.05 respectively) and $OC_{PM2.5}$ (-0.65 %, -0.05 and -0.03 respectively) mass concentrations (see Table 4). According to these tests, SOA production due to the hydrophilic partitioning of the various $VB_{k,i}$ of the VBS-GECKO parameterization is expected to be a minor process.

### 4.2. Sensitivity test to gaseous aging rates and OH radical concentrations

In the VBS-GECKO parameterization, the same rate constant is set for the $VB_{k,i}$ reactions with OH ($k_{OH}$ = $4.0\times10^{-11}$ cm$^3$ molec$^{-1}$ s$^{-1}$). Timescale for gaseous aging is therefore driven by the OH concentrations simulated by the CTM. Simulated OH concentrations depend on the gas-phase chemical mechanisms used in the CTM, with differences on OH concentrations reaching up to 45% between mechanisms (Sarwar et al., 2013). Two simulations were run with modified $k_{OH}$ to examine the sensitivity of SOA production to the rate of chemical aging:

- **$k_{OH}$-VBS-GECKO-low**. In this model configuration, the $VB_{k,i}$+OH rate constants are divided by a factor 2, i.e. $k_{OH}^{low}$ = $2.0\times10^{-11}$ cm$^3$ molec$^{-1}$ s$^{-1}$.

- **$k_{OH}$-VBS-GECKO-high**. In this model configuration, the $VB_{k,i}$+OH rate constants are multiplied by a factor 2, i.e. $k_{OH}^{high}$ = $8.0\times10^{-11}$ cm$^3$ molec$^{-1}$ s$^{-1}$.

The relative difference on the simulated mean OA concentrations between $k_{OH}$-VBS-GECKO-low (respectively $k_{OH}$-VBS-GECKO-high) and ref-VBS-GECKO is given Fig. 6.a (respectively Fig. 6.b). A slight variation of simulated OA concentrations is found (lower than ±10%), with simulated OA concentrations decreasing with the decrease of aging rates and *vice versa*. This result highlights that the gas-phase aging of volatility bins in the VBS-GECKO parameterization promotes functionalization (formation of less volatile bins) rather than fragmentation (formation of more volatile bins), as already shown with tests conducted in box model (Lannuque et al., 2018). The highest relative differences are located over the Mediterranean Sea and North Africa, i.e. areas showing high OH and low OA concentrations (below 4 µg m$^{-3}$, see Fig. 2.a). The $k_{OH}$-VBS-GECKO-high configuration improves statistics, due to an overall increase of the simulated OA concentrations (and contrariwise for the $k_{OH}$-VBS-GECKO-low configuration) (see Table 4). However, the sensitivity of SOA to the gas-phase aging of the VBS-GECKO volatility bins remains weak and aging rates is likely not a major source of uncertainty.

## 4.3. Sensitivity test to the $NO_x$ regime

Similar to the OH discussion above, simulated $HO_2$ and NO concentrations in CTMs are linked to the gas-phase chemical mechanism used. The concentrations of these two species determine the value of the RRR ratio and therefore drive the aging of the various $VB_{k,i}$ (Lannuque et al., 2018). A sensitivity test was performed to examine the sensitivity of the simulated OA to the chemical regime. $HO_2$ or NO concentrations can hardly be modified without changing all the simulation conditions. Here, two simulations were run modifying the $k_{RO2+HO2}$ value used to calculate the RRR:

- **RRR-VBS-GECKO-low**. In this model configuration, RRR ratio is calculated with $k_{RO2+HO2}$ multiplied by 2, i.e. $k_{RO2+HO2}^{RRRlow} = 4.4\times10^{-11}$ cm$^3$ molec$^{-1}$ s$^{-1}$.
- **RRR-VBS-GECKO-high**. In this model configuration, RRR ratio is calculated with $k_{RO2+HO2}$ divided by 2, i.e. $k_{RO2+HO2}^{RRRhigh} = 1.1\times10^{-11}$ cm$^3$ molec$^{-1}$ s$^{-1}$.

Figure 7 presents the mean RRR ratio during the two-month period for both RRR-VBS-GECKO-low (Fig. 7.a) and RRR-VBS-GECKO-high model configurations (Fig. 7.b). The entire range of RRR ratio (from remote $NO_x$ conditions to high $NO_x$ conditions) is covered over Europe with the both model configuration. As expected, the urban, industrial and intense shipping transport areas such as Paris, the Channel, Benelux, northern Italy or Moscow are systematically in the high $NO_x$ regime (RRR close to 1) whereas remote areas over the seas (away from shipping tracks) are systematically in the remote $NO_x$ regime (RRR close to 0). Between these two extremes, the RRR ratio depends on the environmental and meteorological conditions at the location and, in this sensitivity study, on the model configuration for the RRR calculation. Current parameterizations for SOA formation only consider two extreme regimes corresponding to a high-$NO_x$ and a low-$NO_x$ condition. Criteria used to define high and low $NO_x$ differ from a study to another one but the parameterizations are usually optimized at $NO_x$ values typical of rural conditions for low $NO_x$ (corresponding to a RRR ratio of ~0.6) and typical of urban conditions for high $NO_x$ (corresponding to a RRR ratio of ~1) (e.g. Hodzic et al., 2014; Lane et al., 2008). The range of RRR between 0.0 and 0.6 is therefore not considered in most of the parameterizations, although substantial changes in SOA formation were found within this range of RRR (Lannuque et al., 2018).

The relative difference on the simulated mean OA concentrations between RRR-VBS-GECKO-low (respectively RRR-VBS-GECKO-high) and ref-VBS-GECKO is given Fig. 8.a (respectively Fig. 8.b). Results show variations of simulated mean OA concentrations smaller than ~15%. In agreement with previous studies, an increase (decrease) of RRR ratio leads to a decrease (increase) of the simulated OA concentrations (e.g. Donahue et al., 2005; Lannuque et al., 2018; Ng et al., 2007). As expected, the variation is weaker over areas having either an RRR ratio close to 0 or 1, the $NO_x$ regime remaining unchanged among the model configurations. The highest relative differences on OA are found over continental rural areas, i.e. areas showing the largest variation of RRR among the model configurations. Large relative differences are also found over the Mediterranean Sea, owing in part to the low simulated OA concentrations. Similar to the $k_{OH}$ sensitivity tests, the RRR-VBS-GECKO-low configuration increases the overall OA concentrations and improves statistical indicators, and

contrariwise for the RRR-VBS-GECKO-high configuration. Sensitivity on RRR values appears weak enough to likely not be a major source of uncertainty for the VBS-GECKO parameterization.

## 4.4. Sensitivity test to volatility and $P^{sat}$

In the explicit GECKO-A simulations used for the VBS-GECKO optimization, the saturation vapor pressure, $P^{sat}$, of secondary organic compounds was estimated using structure activity relationships (SAR) (see Lannuque et al., 2018). Estimated $P^{sat}$ can typically vary within one order of magnitude according to the SAR used (e.g. Valorso et al., 2011). Two simulations were run to examine the sensitivity of SOA to the uncertainties in $P^{sat}$:

- **$P^{sat}$-VBS-GECKO-low.** In this model configuration, the nominal $P^{sat}$ values of $VB_{k,i}$ are divided by 10.
- **$P^{sat}$-VBS-GECKO-high.** In this model configuration, the nominal $P^{sat}$ values of $VB_{k,i}$ are multiplied by 10.

As OA concentration directly contributes to the partitioning, these two simulations can also be considered as a sensitivity test to the simulated OA concentrations.

The relative difference on the simulated mean OA concentrations between $P^{sat}$-VBS-GECKO-low (respectively $P^{sat}$-VBS-GECKO-high) and ref-VBS-GECKO is given Fig. 9.a (respectively Fig. 9.b). Shifting the volatility of the $VB_{k,i}$ by one order of magnitude leads to an overall change in the simulated mean OA concentrations of about -25% (+25%) when $P^{sat}$ is increased (decreased). A weaker sensitivity is observed over urban areas, such as Paris or Moscow. This behavior is mainly linked to the simulated volatility of OA in the ref-VBS-GECKO simulations. Figure 10 shows the mean volatility of OA over Europe for the reference configuration. Simulated OA contributors are mainly low volatile species (with mean $P^{sat}_{298K}$ between $10^{-10}$ and $10^{-14}$ atm), the highest values being found over urban areas (less aged OA), and the lowest values found over areas close to the boundaries of the domain (linked to a boundary effect in the model). A shift in volatilities over these two types of site has a lower impact on OA concentrations, as OA mean volatilities being either too high (mean $P^{sat}_{298K} \approx 10^{-10}$ atm, upon urban areas) or too low (mean $P^{sat}_{298K} \approx 10^{-14}$ atm, upon boundary areas) for a change in $P^{sat}$ to substantially impact the partitioning. The largest effect is typically observed over central Europe where OA contributors show intermediate mean volatilities (mean $P^{sat}_{298K} \approx 10^{-12}$ atm).

Statistically, the $P^{sat}$-VBS-GECKO-low configuration is the only configuration matching the performance goal for all the simulated OA concentrations ($OC_{PM2.5}$ and $OM_{PM1}$) (see Table 4). For $OC_{PM2.5}$, RMSE is however higher than in reference configuration. Simulated OA concentrations appear to be sensitive to uncertainties in the estimated saturation vapor pressures of the numerous OA contributors considered during the development of the VBS-GECKO parameterization.

## 5. Sensitivity to IVOC emission fluxes from traffic and transport sources

IVOCs have been shown to be a substantial source of SOA in the plume of megacities (e.g. Hodzic et al., 2010; Tsimpidi et al., 2010). Even if several recent studies have been performed to identify the IVOC speciation of different individual

emission sources (e.g. Akherati et al., 2019; Grieshop et al., 2009; Hatch et al., 2018; Jathar et al., 2017; Louvaris et al., 2017; Lu et al., 2018; May et al., 2013a, 2013b, 2013c; Woody et al., 2016), a comprehensive inventory is still not available to represent IVOC emissions by activity sector (gathering several individual emission sources). A large fraction of these IVOCs is thus still not considered in emission inventories. In this section, only IVOC emissions from traffic and transportation sources are treated. Robinson et al. (2007) assumed that IVOC emissions for small off-road diesel engines were equal to 150% of POA emissions, consistent with the Schauer et al. (1999) emission data for 1995 medium-duty diesel vehicles. Recent studies have measured IVOC emissions from (i) exhausts of light-duty gasoline vehicles and (ii) exhausts of both heavy-duty and medium-duty diesel vehicles (Zhao et al., 2015, 2016). Experiments on gasoline exhausts were processed on 42 vehicles and experiments on diesel vehicles on 6 vehicles, the selected vehicles being representative of the transportation fleet in North America. In both cases, Zhao et al. (2015, 2016) have shown that a stronger correlation can be found between IVOC and NMVOC emissions ($R^2$ equal to 0.92 and 0.98 for gasoline and diesel exhausts, respectively) than between IVOC and POA emissions ($R^2$ equal to 0.76 and 0.61 for gasoline and diesel exhausts, respectively). Zhao et al. (2015, 2016) have estimated that IVOC emissions represent about 4% of NMVOC emissions in cold-start cycle to about 16% in hot-start cycle for light-duty gasoline vehicles, and about $60 \pm 10$ % of NMVOC emissions for heavy-duty and medium-duty diesel vehicles.

In this study, the VBS-GECKO parameterization was used to examine the sensitivity of SOA to IVOC emissions from road traffic (SNAP 7) and other mobile sources and machineries (SNAP 8). The following five model configurations, based on different IVOC emission fluxes, were designed for that purpose:

- **IVOC$_{150POA}$.** In this model configuration, IVOC emissions are set to 150% of the semi-volatile POA emissions, based on Robinson et al. (2007).

- **IVOC$_{4VOC}$.** In this configuration, IVOC emissions are set to 4% of NMVOC emissions, based on Zhao et al. (2016) for gasoline vehicles in cold-start cycle.

- **IVOC$_{16VOC}$.** In this configuration, IVOC emissions are set to 16% of NMVOC emissions, based on Zhao et al. (2016) for gasoline vehicles in hot-start cycle.

- **IVOC$_{30VOC}$.** In this configuration, emissions are set to 30% of NMVOC emissions, assuming a mixing of diesel and gasoline vehicle fleets.

- **IVOC$_{65VOC}$.** In this configuration, IVOC emissions are set to 65% of NMVOC emissions, based on Zhao et al. (2015) for diesel vehicles.

As in the reference model configuration, POA are considered as SVOC in these sensitivity tests for traffic and transport emissions. Primary SVOCs and IVOCs (S/IVOCs) constitute a complex mixture of linear, branched and cyclic alkanes, alkenes and aromatics (Fraser et al., 1997; Gentner et al., 2012; Lu et al., 2018; Schauer et al., 1999, 2002). The molecular composition of S/IVOCs emitted in the atmosphere by fossil fuel combustion is however still poorly documented. S/IVOCs at emission were thus considered to be distributed into the 9 volatility bins given by Robinson et al. (2007), with the

provided fraction of primary SVOCs in each SVOC volatility bin, and of estimated primary IVOCs in each IVOC volatility bin. The VBS-GECKO parameterizations for $C_{14}$, $C_{18}$, $C_{22}$ and $C_{26}$ 1-alkenes and n-alkanes were used as surrogate mechanisms for S/IVOCs ($C_{14}$ and $C_{18}$ for IVOCs and $C_{18}$, $C_{22}$ and $C_{26}$ for SVOCs). The $C_{14}$ to $C_{26}$ VBS-GECKO's n-alkanes and 1-alkenes were distributed according to their volatility into the 9 volatility bins of Robinson et al. (2007).

Correspondences are shown in Figure 11, for the example of the $IVOC_{150POA}$ model configuration. The distribution of alkanes and alkenes was estimated based on (i) the EMEP guidebook (https://www.eea.europa.eu/publications/emep-eea-guidebook-2016), providing speciation data for emissions for various types of vehicles and (ii) the COPERT4 software (Ntziachristos et al., 2009) providing data for a vehicle fleet. Data are only available for light compounds and are here extrapolated to the heavy ones for the needs of the study. Thus, 75 % of the primary S/IVOCs are here assumed to be alkanes

and 25 % alkenes. The primary SVOC total emissions and distributions over volatility bins are unchanged between each simulation. The distribution of IVOCs among volatility bins is also unchanged but the total IVOC emissions are modulated according to the 5 IVOC emission scenarios described before (i.e. $IVOC_{150POA}$, $IVOC_{4VOC}$, $IVOC_{16VOC}$, $IVOC_{30VOC}$ and $IVOC_{65VOC}$). Table 5 gives the speciation of VBS-GECKO species for the various model configurations and the VBS-GECKO mechanism for S/IVOCs is presented in Table S3 of supplementary material.

Figure 12 shows the mean OA mass concentrations simulated for the 5 IVOC emission configurations, and the absolute and relative differences with the ref-VBS-GECKO simulation without IVOC emissions. Table 6 presents the statistical results calculated on daily averaged concentrations (RMSE, Pearson's correlation coefficient, MFE and MFB) for the different IVOC emission configurations, and their difference with those of the ref-VBS-GECKO configuration. As discussed previously, the highest concentrations are simulated over northern Italy (see Fig. 2). For this area, accounting for IVOC

emissions increases the simulated concentrations of OA up to 3 µg m$^{-3}$ with the $IVOC_{65VOC}$ model configuration. As expected, OA concentration increases when IVOC emissions over Europe are taken into account during the simulated period, with an overall mean increase of about 12, 2, 5, 10 and 20% for the $IVOC_{150POA}$, $IVOC_{4VOC}$, $IVOC_{16VOC}$, $IVOC_{30VOC}$ and $IVOC_{65VOC}$ configuration, respectively. The relative differences show large increases of OA concentrations (reaching +40%) over a wide area including North Sea and Benelux for the $IVOC_{65VOC}$ configuration, owing to the low simulated OA

concentrations with the ref-VBS-GECKO configuration. The $IVOC_{150POA}$ configuration leads to mean OA mass concentrations lying between the $IVOC_{16VOC}$ and the $IVOC_{30VOC}$ configurations. Area showing substantial changes in simulated OA are however different between these model configurations. In the $IVOC_{150POA}$ configuration, the largest OA concentration increase is simulated over the Channel and Gibraltar's Detroit (up to +80%). These results were expected for this model configuration based on POA emissions. Indeed, ships are one of the most important sources of POA but emit a

relatively small amount of NMVOCs. For example, the EMEP inventory for 2013 estimates an average NMVOC/POA emission ratio of ~4 for road traffic in Europe and ~0.4 for shipping in the studied domain.

Taking into account SOA formation from IVOC precursors improves the statistical indicators for the simulated concentrations of $OM_{PM1}$. As discussed previously, the ref-VBS-GECKO configuration underestimates $OM_{PM1}$. Including IVOC emission increases the mean $OM_{PM1}$ concentrations at the stations of about 5, 2, 5, 7 or 13% for the $IVOC_{150POA}$,

IVOC$_{4VOC}$, IVOC$_{16VOC}$, IVOC$_{30VOC}$ or IVOC$_{65VOC}$ configuration, respectively. Increasing IVOC emissions provide better statistical indicators for OM$_{PM1}$, with MFE and MFB significantly closer to the performance goal (MFE decreases by 0.06 and |MFB| decreases by 0.09 between IVOC$_{4VOC}$ and IVOC$_{65VOC}$ configurations, see Table 6). For OC$_{PM2.5}$ however, the opposite trend is observed with a degradation of the statistical indicators (Table 6). The ref-VBS-GECKO configuration

leads to a slight overestimation of OC$_{PM2.5}$ concentrations over some stations (e.g. Iskrba, see Fig. 3) and adding the SOA source from IVOCs strengthens the deviation (up to about +30% of RMSE for the IVOC$_{65VOC}$ configuration), even if the correlation is not significantly modified.

IVOC oxidation appears to be a significant SOA source at some locations (e.g. the Cabauw station), especially in the IVOC$_{30VOC}$ and IVOC$_{65VOC}$ configuration. However, the resulting OA increase remains too weak to fill in the gaps between

observations and simulated data (maximum increase around +40%). For example, time series presented in Figure 13 show that adding IVOC emission increases systematically the simulated OA concentrations, but not enough to explain the OA peaks recorded at the anthropogenic stations (see Fig. 13.b). Moreover, accounting for IVOC emission strengthens the disagreement of the simulated concentrations with observations over other areas (e.g. at Iskrba station).

The various IVOC emission configurations are aimed to answer to the question: with constant POA and NMVOC emissions

for the traffic, does IVOC emissions typical of diesel vehicles (upper limit) or gasoline vehicles (lower limit) significantly change OA concentrations in Europe, and in particular in anthropogenic areas? At a local scale where anthropogenic sources are dominant, IVOC emissions from traffic and transportation sources appear to be a significant source of OA and simulated OA concentrations are dependent to the IVOC emission configuration ($\sim$+3 µg m$^{-3}$ in northern Italy for IVOC$_{65VOC}$ against IVOC$_{4VOC}$). At a continental scale outside anthropogenic areas, the low variations observed on simulated OA concentrations

between the different IVOC emission configurations suggest that IVOCs from traffic and transportation sources are likely not a major source of SOA.

## 6. Tracking OA sources

Apportionment of OA sources is investigated in this section. The study takes into account OA formation from IVOC oxidation and is based on the IVOC$_{30VOC}$ model configuration. Figure 14 shows the contribution of the various OA sources

to the simulated OA concentrations during the July-August 2013 period and the mean daily profiles at two stations located in areas dominantly impacted by anthropogenic air masses (Cabauw and Palaiseau) and two stations located in areas dominantly impacted by biogenic air masses (Birkenes II and Iskrba). SOA constitutes the main fraction of OA whatever the environment. This secondary fraction typically grows from anthropogenic impacted areas (about 70 % at Palaiseau station) to remote areas (about 95 % at Iskrba station). This trend is in agreement with what is usually observed or simulated for

summertime periods (e.g. Aksoyoglu et al. 2011; Belis et al., 2013). For the remote stations Birkenes II and Iskrba, respectively 82 and 67% of the simulated OA concentration comes from a biogenic source. Contrariwise, anthropogenic

sources are the major OA contributors at anthropogenic impacted stations (65 and 60% of OA at the Cabauw and Palaiseau stations, respectively.

Among OA biogenic sources, terpene oxidation is clearly found as the major contributor of OA during the summer period, contributing from 35% (at anthropogenic impacted stations) to 80% (at remote stations) of the total OA mass. The 60% increase of OA mass concentration observed in north Europe between $H_2O$ and VBS-GECKO parameterizations (see Fig. 2) is also mainly related to SOA formation from terpene, especially ocimene and limonene. In our simulation, SOA produced by isoprene oxidation does not represent a substantial fraction of OA at the selected measurement stations. The major contribution of isoprene SOA to OA reaches about 5% (see Fig. 14.h) and is observed at the Iskrba station during diurnal conditions.

The anthropogenic fraction of OA is found to be dominated by residential biomass burning sources (BBOA). Indeed, according to the temporal factors used in CHIMERE (based on GENEMIS, Ebel et al., 1997; Friedrich, 2000), 4% of annual emissions of residential BBOA occurs during July-August, leading to a non-negligible amount of residential BBOA during summer. This result remains however subject to caution, owing to the large uncertainties in the temporalization of biomass burning emissions in the model. The primary organic fraction (i.e. condensed primary SVOCs) from traffic emissions is found to be substantial in the OA budget only at night in urban areas. On the other hand, the secondary organic fraction produced by traffic emissions can represent about 50% of diurnal anthropogenic OA at stations near urban areas (i.e. Palaiseau and Cabauw). OA formed by the oxidation of mono-aromatic species is found to be negligible over Europe (less than 0.025 µg m$^{-3}$ on average over the studied domain). Figure 15 shows the contribution of traffic emission to the simulated OA concentrations for the July-August 2013 period for 3 categories of precursors: SVOCs, IVOCs and mono-aromatic compounds. As mentioned above, the OA concentrations from mono-aromatic compound oxidation are negligible compared to concentrations from traffic S/IVOC oxidation. Globally, in our study over Europe, OA concentrations produced from traffic S/IVOC oxidation are of the same order of magnitude. OA from primary SVOCs is locally more important close to sources (i.e. Northern Italy, Moscow, Paris, Gibraltar, etc). OA from IVOC is globally higher far away from the sources, with a higher dispersion over Europe (Fig. 15). This higher dispersion is expected owing to the larger timescale required to produce low volatility species via multistep oxidation processes in the plumes of high emission area.

The distributions of OA within the volatility bins (given in Figure S2 of supplementary material) show similar features from one station to another. The results suggest that OA over Europe has relatively low volatility during summertime. Indeed, the VBS-GECKO contributors to OA have very low volatility: ~80% of the OA contributors from VBS-GECKO are volatility bins 7 to 5 ($VB_{k,7}$, $VB_{k,6}$ and $VB_{k,5}$ species), i.e. having saturation vapor pressure at 298 K of $10^{-14}$, $10^{-12}$ and $10^{-11}$ atm respectively.

## 7. Conclusions

The VBS-GECKO parameterization for SOA production was developed based on explicit mechanisms generated with the GECKO-A tool. The VBS-GECKO parameterization was fitted using box modelling results for a selected set of parent compounds including terpenes, mono-aromatic compounds, linear alkanes and alkenes and for various environmental conditions, including different $NO_x$ regimes, temperatures, OA loads (Lannuque et al., 2018). In this study, the VBS-GECKO parameterization was evaluated in the CHIMERE β 2017 CTM over Europe during summertime.

The VBS-GECKO parameterization shows good performances to simulate OA concentrations over Europe in the summer. Calculated mean fractional biases and mean fractional errors on $PM_{2.5}$, $OC_{PM2.5}$ and $OM_{PM1}$ satisfy the performance criteria of Boylan and Russel (2006). The model configuration including the VBS-GECKO parameterization yields to higher OA concentrations compared to the former reference configuration including the $H^2O$ parameterization. The deviations between the two configurations are especially marked over northern Europe, with an increase factor of ~60%. Outside this area, the OA increases obtained with the VBS-GECKO configuration are slight. Statistically, the use of the VBS-GECKO improves the overall MFB, MFE and RMSE and does not modify significantly correlation coefficients. Tests performed to examine the sensitivity of simulated OA concentrations to hydro-solubility, volatility, aging rates and $NO_x$ regimes have shown that the VBS-GECKO parameterization provides consistent results that are not subject to large deviations induced by parameters provided by the gas phase mechanism included in the CTM (e.g. $HO_x$ or $NO_x$ concentrations). However, the OA concentrations remain underestimated with the VBS-GECKO model configuration, especially in areas with a significant contribution of anthropogenic sources (e.g. reaching a factor of 2.5 for $OC_{PM2.5}$ at the NL0644R station in Netherlands). None of the conducted sensitivity test leads to OA variations large enough to fill the gaps between measurements and simulated concentrations at the anthropogenic stations.

The analysis of simulated OA shows that, during summertime, the main fraction is made of secondary matter which represents ~85% of the total mean OA concentration. A large fraction of the simulated OA comes from biogenic sources (between 30 and 85% of the total OA), especially from terpene oxidation which represents ~95% of these biogenic sources. For the conditions examined in this study, OA formed by the oxidation of mono-aromatic compounds appears to be negligible with maximum mean concentrations of 0.025 µg m$^{-3}$ over North Sea and Benelux. Note that ignoring SOA production from these precursors in the model would substantially reduce the number of $VB_{k,i}$ species currently considered in the VBS-GECKO parameterization. The simulated OA was found to be made of species having low and extremely low volatilities in remote areas, but also of SVOCs closer to major anthropogenic sources.

Finally, IVOC oxidation was added to examine the contribution of this additional source to the SOA budget. Five model configurations with distinct IVOC emissions from traffic were tested and compared using the VBS-GECKO parameterization in CHIMERE. As expected, considering the emission of IVOCs by traffic and transport sources was found to globally increase background OA concentrations. Although SOA production from traffic IVOC oxidation can locally be significant (up to ~+3 µg m$^{-3}$ in northern Italy, assuming IVOC emissions represents 65% of NMVOC emissions), this

additional OA source remains too small to explain the gap between simulated and measured values at stations where anthropogenic sources are dominant. This first application of this new VBS-GECKO parameterization has been shown to provide consistent results. This outcome creates motivation to extend the exploration to wintertime conditions, and expand the list of parent compounds considered, in particular to include SOA formation from oxidation of isoprene, sesquiterpenes

or organics species emitted by residential biomass burning, a prerequisite to extend the evaluation and analysis to wintertime when this source is dominant. This is the subject of ongoing studies. The VBS-GECKO is a heavy parameterization in term of species number. Calculation time is multiplied by two using the complete VBS-GECKO scheme with IVOCs compared to $H^2O$. This study has shown that the number of species can be optimized. For example, because of the low influence on OA concentrations, the representation of the SOA formed by the oxidation of mono-aromatic species can be highly simplified

and $C_{10}$ precursors even removed.

**Data availability**

Daily averages and mean day profiles for the 17 model configurations presented in this article have been made available on Zenodo: https://zenodo.org/record/ 1654297 / (last access: 29 November 2018).

**Competing interests**

The authors declare that they have no conflict of interest.

**Author Contribution**

VL implemented the parameterization in the air quality model and conducted the simulations and the sensitivity tests. All the authors contributed to design the research, to interpret the data and to write the article.

**Acknowledgments**

The authors gratefully acknowledge Wenche Aas, Olivier Favez, Liine Heikkinen and Laurent Poulain for providing ACSM data realized in the framework of ACTRIS. This work was financially supported by the French Environment and Energy Management Agency (ADEME) and INERIS. We also thank French Ministry of Ecology for its financial support. Simulations were performed using the TGCC-CCRT super computers.

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

**Table 1 – List of the VBS-GECKO species and associated properties.**

| Species | Partition[b] | MW (g.mol$^{-1}$) | $P^{sat}_{298K}$ (atm) | $H^{eff}_{298K}$ (mol.L$^{-1}$.atm$^{-1}$) | $\Delta H_{vap}$ (kJ.mol$^{-1}$) |
|---|---|---|---|---|---|
| α-pinene | | 136 | $4.47 \times 10^{-3}$ | $1.70 \times 10^{-2}$ | 46 |
| β-pinene | | 136 | $4.79 \times 10^{-3}$ | $1.70 \times 10^{-2}$ | 45 |
| Limonene | | 136 | $4.57 \times 10^{-3}$ | $1.70 \times 10^{-2}$ | 46 |
| Benzene | | 78 | $6.61 \times 10^{-2}$ | 0.21 | 36 |
| Toluene | | 92 | $2.14 \times 10^{-2}$ | 0.18 | 40 |
| O-xylene | | 106 | $7.24 \times 10^{-3}$ | 0.25 | 45 |
| M-xylene | | 106 | $5.75 \times 10^{-3}$ | 0.16 | 45 |
| P-xylene | | 106 | $7.24 \times 10^{-3}$ | 0.17 | 45 |
| decane | | 142 | $1.90 \times 10^{-3}$ | $1.41 \times 10^{-4}$ | 50 |
| Tetradecane | * | 198 | $3.63 \times 10^{-5}$ | $7.94 \times 10^{-5}$ | 66 |
| Octadecane | * | 254 | $7.76 \times 10^{-7}$ | $2.57 \times 10^{-5}$ | 83 |
| Docosane | * | 310 | $1.66 \times 10^{-8}$ | $8.51 \times 10^{-6}$ | 100 |
| Hexacosane | * | 366 | $3.39 \times 10^{-10}$ | $2.82 \times 10^{-6}$ | 118 |
| Decene | | 140 | $2.04 \times 10^{-3}$ | $1.00 \times 10^{-3}$ | 50 |
| Tetradecene | * | 196 | $3.98 \times 10^{-5}$ | $3.31 \times 10^{-4}$ | 66 |
| Octadecene | * | 253 | $8.71 \times 10^{-7}$ | $1.10 \times 10^{-4}$ | 82 |
| Docosene | * | 308 | $1.91 \times 10^{-8}$ | $3.63 \times 10^{-5}$ | 99 |
| Hexacosene | * | 364 | $4.07 \times 10^{-10}$ | $1.20 \times 10^{-5}$ | 117 |
| $VB_{k,1}$[a] | * | 210 | $3.16 \times 10^{-7}$ | $1.0 \times 10^{6}$ | 90 |
| $VB_{k,2}$[a] | * | 240 | $1.0 \times 10^{-8}$ | $1.0 \times 10^{7}$ | 105 |
| $VB_{k,3}$[a] | * | 270 | $1.0 \times 10^{-9}$ | $1.0 \times 10^{8}$ | 115 |
| $VB_{k,4}$[a] | * | 300 | $1.0 \times 10^{-10}$ | $1.0 \times 10^{9}$ | 125 |
| $VB_{k,5}$[a] | * | 330 | $1.0 \times 10^{-11}$ | $1.0 \times 10^{10}$ | 135 |
| $VB_{k,6}$[a] | * | 360 | $1.0 \times 10^{-12}$ | $1.0 \times 10^{11}$ | 145 |
| $VB_{k,7}$[a] | * | 390 | $1.0 \times 10^{-14}$ | $1.0 \times 10^{12}$ | 165 |

[a] Properties of the bins do not depend on the precursor k (see Lannuque et al., 2018)
[b] Gas/particle partitioning is implemented in CHIMERE for species with a * only

**Table 2 – Distribution of the NMVOC emission in the VBS-GECKO species.**

| VBS-GECKO | Emitted NMVOC in CHIMERE (Passant, 2002) |
|---|---|
| Decane | C10 alkanes; C10 cycloalkanes; 75% of C11 alkanes; 50% of C12 alkanes; 25% of C13 alkanes |
| Tetradecane | 25% of C11 alkanes; 50% of C12 alkanes; 75% of C13 alkanes |
| Decene | C10 alkenes |
| Benzene | Benzene |
| Toluene | 25% of C9, C10, C13 and unspeciated aromatic hydrocarbons, ethylbenzene, isopropylbenzene, propylbenzene, phenol, toluene and styrene |
| O-xylene | 25% of C9, C10, C13 and unspeciated aromatic hydrocarbons; 2-ethyltoluene; indan; 33% of ethyltoluene; 33% of methylpropylbenzene; o-xylene |
| M-xylene | 25% of C9, C10, C13 and unspeciated aromatic hydrocarbons, tetramethylbenzene; trimethylbenzene; 1-methyl-3-isopropylbenzene; 3-ethyltoluene; ethyldimethylbenzene; 33% of ethyltoluene; 33% of methylpropylbenzene; m-xylene |
| P-xylene | 25% of C9, C10, C13 and unspeciated aromatic hydrocarbons; 1-methyl-4-isopropylbenzene; 4-ethyltoluene; 33% of ethyltoluene; 33% of methylpropylbenzene; p-xylene |

**Table 3 – Statistical results calculated on daily averaged concentrations for the ref-VBS-GECKO simulations and differences between ref-VBS-GECKO and H²O statistical indicators.**

| | Observations | | VBS-GECKO results | | | | | Differences with H²O results | | | | |
|---|---|---|---|---|---|---|---|---|---|---|---|---|
| | Number | Mean | Mean | RMSE | r | MFB | MFE | Δ mean | Δ RMSE | Δ r | Δ MFB | Δ MFE |
| | - | (µg.m⁻³) | (µg.m⁻³) | (µg.m⁻³) | - | - | - | (%) | (%) | - | (dist from 0) | - |
| PM$_{2.5}$ | 2002 | 9.07 | 7.65 | 6.14 | 0.42 | -0.09 | 0.37 | +5.52 | -1.76 | +0.01 | -0.05 | -0.01 |
| OM$_{PM1}$ | 237 | 3.24 | 1.93 | 2.28 | 0.79 | -0.47 | 0.57 | +31.3 | -11.6 | -0.04 | -0.25 | -0.17 |
| OC$_{PM2.5}$ | 235 | 2.52 | 2.59 | 1.75 | 0.57 | -0.16 | 0.51 | +15.6 | +5.42 | +0.02 | -0.17 | -0.08 |

**Table 4 – Statistical results calculated on daily averaged concentrations simulated with the various model configurations and differences with the ref-VBS-GECKO configuration statistical indicators given Table 3.**

| Model configuration | | Sensitivity test results | | | | | Differences to ref-VBS-GECKO results | | | | |
|---|---|---|---|---|---|---|---|---|---|---|---|
| | | Mean | RMSE | r | MFB | MFE | Δ mean | Δ RMSE | Δ r | Δ MFB | Δ MFE |
| | | (µg.m⁻³) | (µg.m⁻³) | - | - | - | (%) | (%) | - | (dist from 0) | - |
| hydro-VBS-GECKO | OM$_{PM1}$ | 1.93 | 2.28 | 0.79 | -0.47 | 0.57 | 0.00 | 0.00 | 0.00 | 0.00 | 0.00 |
| | OC$_{PM2.5}$ | 2.59 | 1.75 | 0.57 | -0.16 | 0.51 | 0.00 | 0.00 | 0.00 | 0.00 | 0.00 |
| hydro-VBS-GECKO-high | OM$_{PM1}$ | 2.07 | 2.17 | 0.79 | -0.41 | 0.52 | +7.43 | -4.58 | 0.00 | -0.06 | -0.05 |
| | OC$_{PM2.5}$ | 2.71 | 1.73 | 0.57 | -0.11 | 0.48 | +4.80 | -0.65 | 0.00 | -0.05 | -0.03 |
| k$_{OH}$-VBS-GECKO-low | OM$_{PM1}$ | 1.87 | 2.32 | 0.78 | -0.49 | 0.59 | -2.70 | +1.90 | -0.01 | +0.02 | +0.02 |
| | OC$_{PM2.5}$ | 2.52 | 1.73 | 0.57 | -0.19 | 0.53 | -2.40 | -0.65 | 0.00 | +0.03 | +0.02 |
| k$_{OH}$-VB-GECKO-high | OM$_{PM1}$ | 2.02 | 2.21 | 0.79 | -0.43 | 0.54 | +4.72 | -2.67 | 0.00 | -0.04 | -0.03 |
| | OC$_{PM2.5}$ | 2.70 | 1.76 | 0.57 | -0.12 | 0.49 | +4.32 | +0.65 | 0.00 | -0.04 | -0.02 |
| RRR-VBS-GECKO-low | OM$_{PM1}$ | 2.11 | 2.13 | 0.80 | -0.41 | 0.52 | +9.45 | -6.48 | 0.01 | -0.06 | -0.05 |
| | OC$_{PM2.5}$ | 2.82 | 1.81 | 0.57 | -0.08 | 0.48 | +9.13 | +3.92 | 0.00 | -0.08 | -0.03 |
| RRR-VBS-GECKO-high | OM$_{PM1}$ | 1.73 | 2.42 | 0.78 | -0.54 | 0.64 | -10.1 | +6.48 | -0.01 | +0.07 | +0.07 |
| | OC$_{PM2.5}$ | 2.36 | 1.72 | 0.56 | -0.25 | 0.56 | -8.65 | -1.30 | -0.01 | +0.09 | +0.05 |
| P$^{sat}$-VBS-GECKO-low | OM$_{PM1}$ | 2.42 | 1.92 | 0.80 | -0.31 | 0.44 | +25.6 | -15.6 | +0.01 | -0.16 | -0.13 |
| | OC$_{PM2.5}$ | 3.17 | 1.99 | 0.58 | 0.02 | 0.45 | +22.5 | +13.7 | +0.01 | -0.18 | -0.06 |
| P$^{sat}$-VBS-GECKO-high | OM$_{PM1}$ | 1.47 | 2.63 | 0.76 | -0.65 | 0.73 | -23.6 | +15.6 | -0.03 | +0.18 | +0.16 |
| | OC$_{PM2.5}$ | 2.07 | 1.76 | 0.56 | -0.36 | 0.63 | -19.7 | +0.65 | -0.01 | +0.20 | +0.12 |

**Table 5 – Distribution of the VBS-GECKO surrogate species for IVOC emission in the various model configurations.**

| Model configuration | Species | | | |
| --- | --- | --- | --- | --- |
| | C14 | C18 | C22 | C26 |
| **IVOC$_{150POA}$** | 105 % of POA | 80 % POA | 40 % of POA | 25 % of POA |
| **IVOC$_{4VOC}$** | 2.8 % of NMVOCs | 35 % of POA + 1,2 % of NMVOCs | 40 % of POA | 25 % of POA |
| **IVOC$_{16VOC}$** | 11.2 % of NMVOCs | 35 % of POA + 4.8 % of NMVOCs | 40 % of POA | 25 % of POA |
| **IVOC$_{30VOC}$** | 21 % of NMVOCs | 35 % of POA + 9 % of NMVOCs | 40 % of POA | 25 % of POA |
| **IVOC$_{65VOC}$** | 45.5 % of NMVOCs | 35 % of POA + 19.5 % of NMVOCs | 40 % of POA | 25 % of POA |

**Table 6 – Statistical results calculated on daily averaged concentrations simulated with VBS-GECKO considering IVOC emissions and differences with those of the ref-VBS-GECKO (without IVOCs) from Table 3.**

| Model configuration | | VBS-GECKO with IVOC results | | | | | Differences with ref-VBS-GECKO | | | | |
| --- | --- | --- | --- | --- | --- | --- | --- | --- | --- | --- | --- |
| | | Mean | RMSE | r | MFB | MFE | Δ Mean | Δ RMSE | Δ r | Δ MFB | Δ MFE |
| | | ($\mu$g.m$^{-3}$) | ($\mu$g.m$^{-3}$) | - | - | - | % | % | - | (dist from 0) | - |
| **IVOC$_{150POA}$** | OM$_{PM1}$ | 2.02 | 2.18 | 0.80 | -0.43 | 0.53 | +4.66 | -4.38 | +0.01 | -0.04 | -0.04 |
| | OC$_{PM2.5}$ | 2.72 | 1.83 | 0.57 | -0.11 | 0.5 | +5.01 | +4.57 | 0.00 | -0.05 | -0.01 |
| **IVOC$_{4VOC}$** | OM$_{PM1}$ | 1.96 | 2.24 | 0.79 | -0.46 | 0.56 | +1.55 | -1.75 | 0.00 | -0.01 | -0.01 |
| | OC$_{PM2.5}$ | 2.64 | 1.79 | 0.57 | -0.15 | 0.51 | +1.93 | +2.28 | 0.00 | -0.01 | 0.00 |
| **IVOC$_{16VOC}$** | OM$_{PM1}$ | 2.01 | 2.20 | 0.8 | -0.44 | 0.54 | +4.14 | -3.50 | +0.01 | -0.03 | -0.03 |
| | OC$_{PM2.5}$ | 2.73 | 1.88 | 0.57 | -0.12 | 0.51 | +5.40 | +7.42 | 0.00 | -0.04 | 0.00 |
| **IVOC$_{30VOC}$** | OM$_{PM1}$ | 2.06 | 2.15 | 0.80 | -0.42 | 0.53 | +6.73 | -5.70 | +0.01 | -0.05 | -0.04 |
| | OC$_{PM2.5}$ | 2.84 | 1.98 | 0.57 | -0.09 | 0.51 | +9.65 | +13.1 | 0.00 | -0.07 | 0.00 |
| **IVOC$_{65VOC}$** | OM$_{PM1}$ | 2.18 | 2.04 | 0.81 | -0.37 | 0.50 | +12.9 | -10.5 | +0.02 | -0.10 | -0.07 |
| | OC$_{PM2.5}$ | 3.10 | 2.29 | 0.56 | -0.02 | 0.50 | +19.6 | +30.8 | -0.01 | -0.14 | -0.01 |

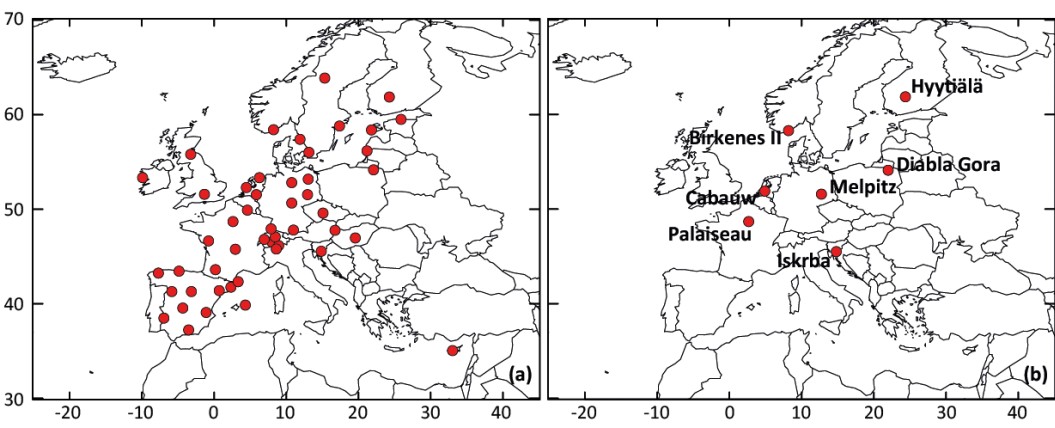

**Figure 1 – Location of rural background stations used for (a) the statistical evaluations and (b) time series comparisons.**

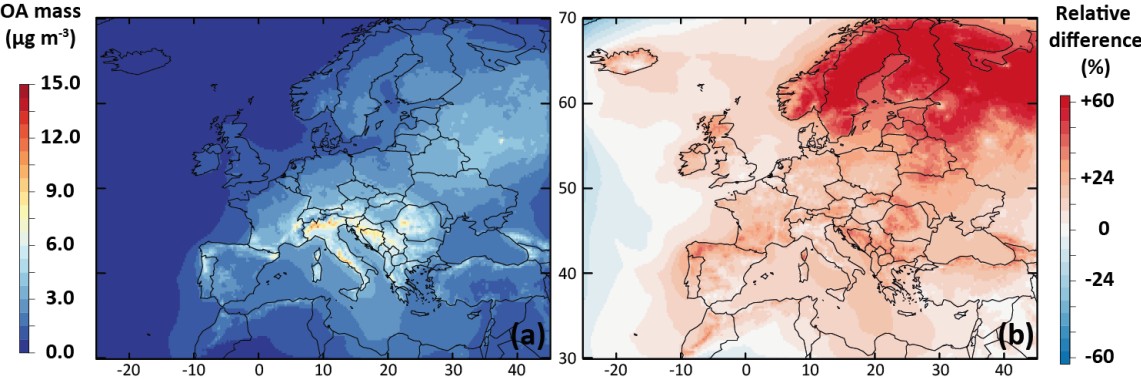

**Figure 2 – Mean OA mass concentrations simulated with the ref-VBS-GECKO model configuration over Europe for the July-August 2013 period (a) and relative difference of the simulated mean OA mass concentrations between the ref-VBS-GECKO and the $H^2O$ configuration (b).**

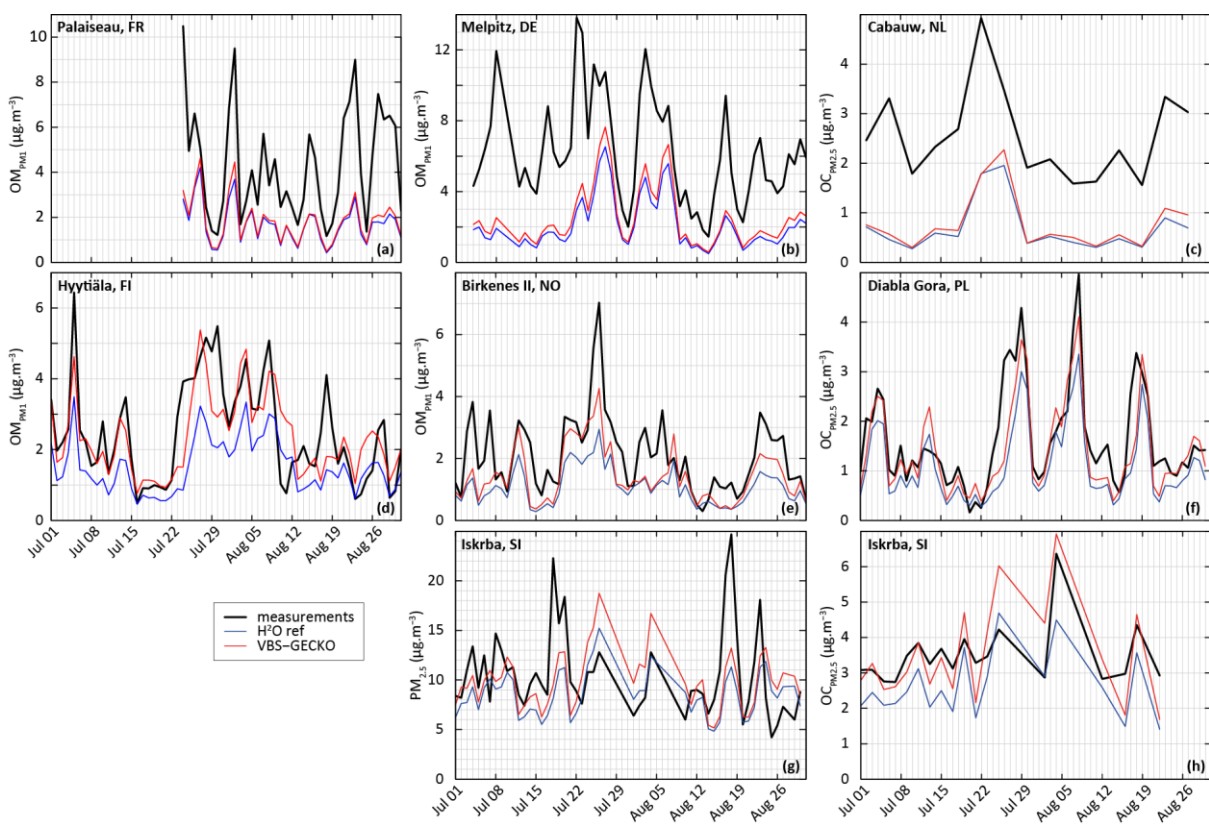

**Figure 3 – Measured (black) and simulated (with $H^2O$ in blue and ref-VBS-GECKO in red) temporal evolution of daily averaged OM$_{PM1}$ concentrations (panels a, b, d and e) OC$_{PM2.5}$ concentrations (panels c, f and h) and PM$_{2.5}$ concentrations (panel g). Top panels are for stations influenced by anthropogenic sources in France (Palaiseau station, a), Germany (Melpitz station, b) and Netherland (Cabauw station, c), middle panels are stations in remote areas in Finland (Hyytiälä station, d), Norway (Birkenes II**

10 **station, d) and Poland (Diabla Gora station, e) and bottom panels are for a station located in a remote area in Slovenia (Iskrba, g and h).**

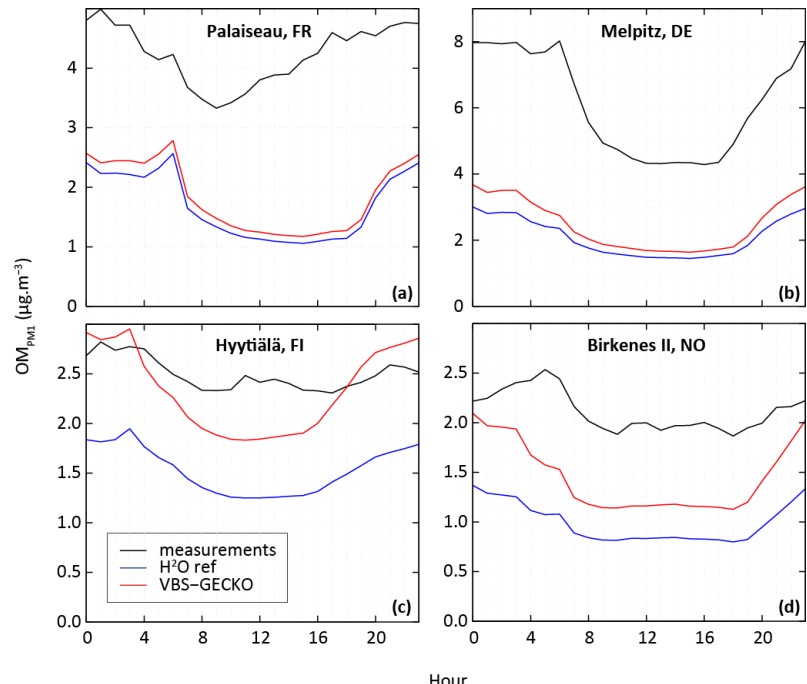

**Figure 4 – Measured (black) and simulated mean diurnal profile (in UTC) with the $H^2O$ model configuration (blue) and the ref-VBS-GECKO model configuration (red) for $OM_{PM1}$ concentration at stations influenced dominantly by anthropogenic sources (Palaiseau (a) and Melpitz (b)) and by biogenic sources (Hyytiälä (c) and Birkenes II (d)).**

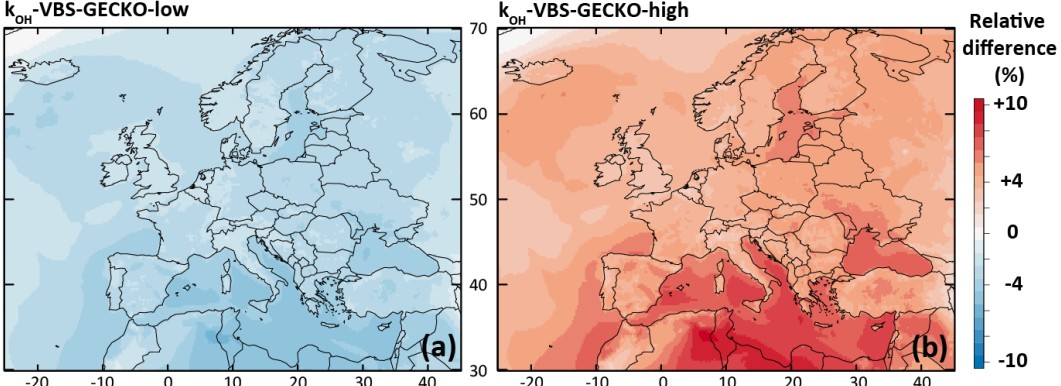

**Figure 5** – Relative differences on the simulated mean OA concentrations between (a) the Hydro-VBS-GECKO and the ref-VBS-GECKO and (b) the Hydro-VBS-GECKO-high and the ref-VBS-GECKO model configurations for the two-month period. Bottom panels represent relative differences on the simulated mean OA concentrations between the Hydro-VBS-GECKO-high and the ref-VBS-GECKO due to variation in deposition (c) or partitioning (d).

**Figure 6** – Relative differences on the simulated mean OA concentrations between (a) the $k_{OH}$-VBS-GECKO-low and the ref-VBS-GECKO model configurations and (b) the $k_{OH}$-VBS-GECKO-high and the ref-VBS-GECKO model configurations for the two-month period.

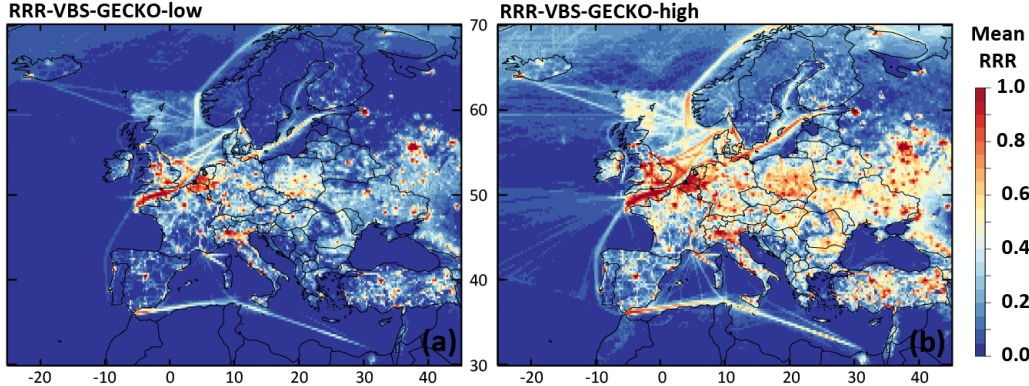

**Figure 7 – Mean RRR over Europe during the two-month period for (a) the RRR-VBS-GECKO-low and (b) the RRR-VBS-GECKO-high model configurations.**

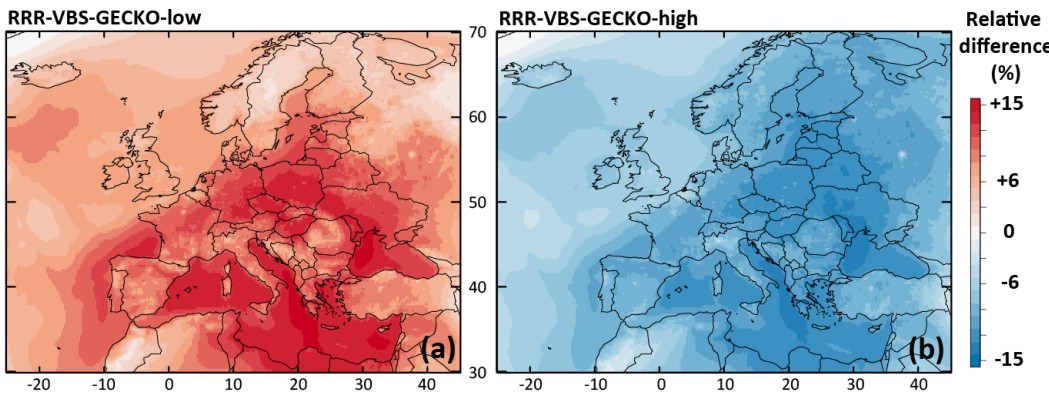

**Figure 8 – Relative differences on the simulated mean OA concentrations between (a) the RRR-VBS-GECKO-low and the ref-VBS-GECKO model configurations and (b) between the RRR-VBS-GECKO-high and the ref-VBS-GECKO model configurations for the two-month period.**

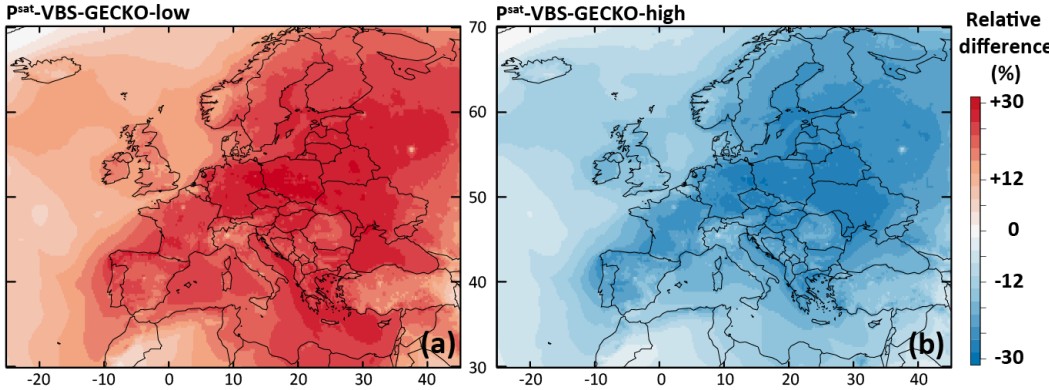

**Figure 9 – Relative difference on the simulated mean OA concentrations between (a) the P$^{sat}$-VBS- -low and the ref-VBS-GECKO model configurations and (b) between the P$^{sat}$-VBS-GECKO-high and the ref-VBS-GECKO model configurations for the two-month period.**

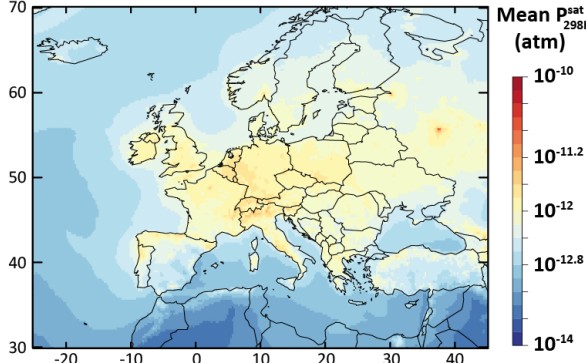

**Figure 10 – Simulated average volatility of OA in term of P$^{sat}_{298K}$ upon Europe during the July-August 2013 period for the ref-VBS-GECKO model configuration.**

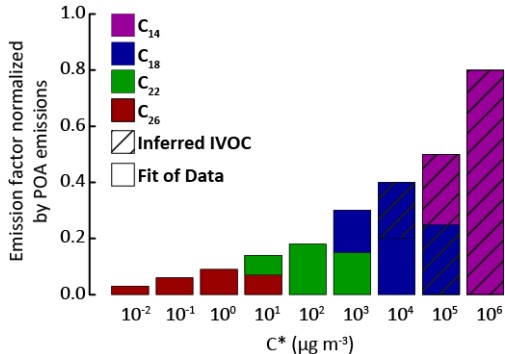

10    **Figure 11 – Distribution of the VBS-GECKO species into the S/IVOC volatility bins of Robinson et al. (2007). Normalized emission factors by POA emissions for IVOCs are used for the IVOC$_{150POA}$ model configuration.**

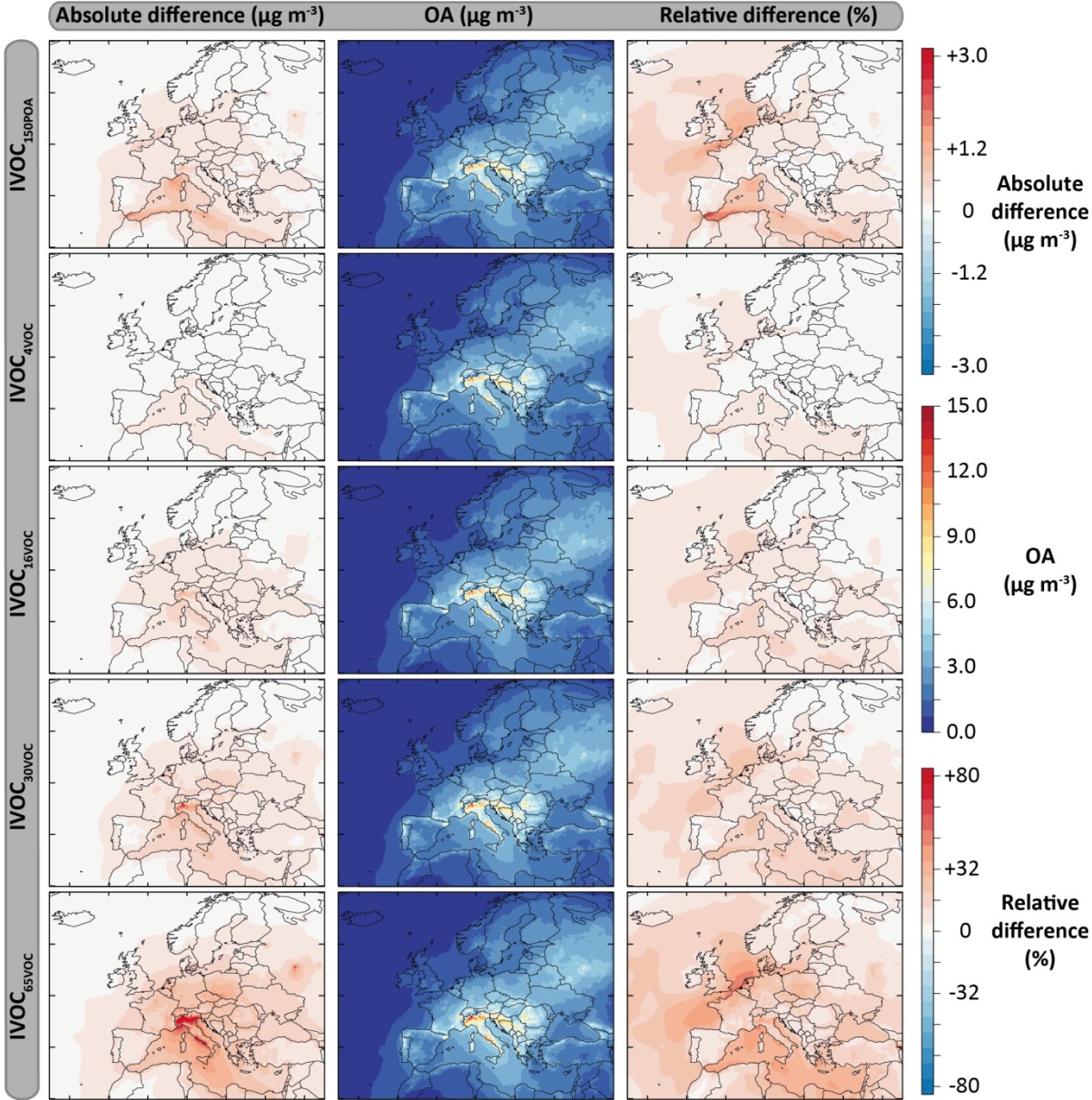

**Figure 12 – Mean OA mass concentrations simulated with the model configurations including the IVOC emissions for the July-August 2013 period over Europe (second column), and absolute and relative differences with the ref-VBS-GECKO model configurations (left and right columns respectively). Results are given for the following model configurations: IVOC$_{150POA}$ (first row), IVOC$_{4VOC}$ (second row), IVOC$_{16VOC}$ (third row), IVOC$_{30VOC}$ (fourth row) and IVOC$_{65VOC}$ (fifth row).**

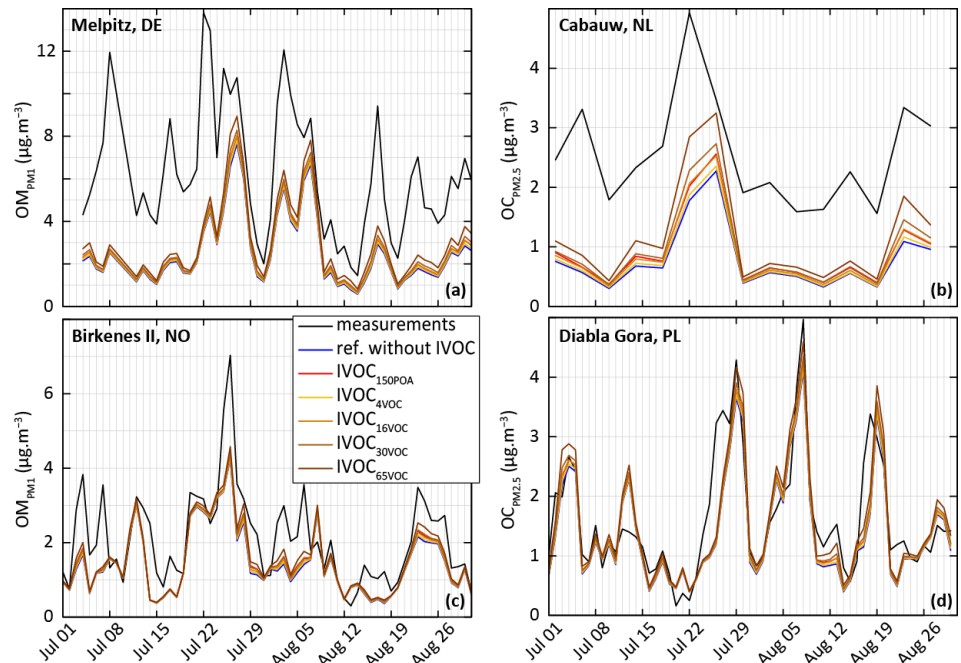

**Figure 13 – Measured and simulated (for the ref-VBS-GECKO configuration without IVOC and the different model configuration considering IVOC emissions) temporal evolution of daily averaged OM$_{PM1}$ concentrations (panels a and c) and OC$_{PM2.5}$ concentrations (panels b and d). Top panels are for stations close to anthropogenic sources in Germany (Melpitz station, a) and Netherland (Cabauw station, b) and bottom panels for stations in remote areas in Norway (Birkenes II station, c) and Poland (Diabla Gora station, d).**

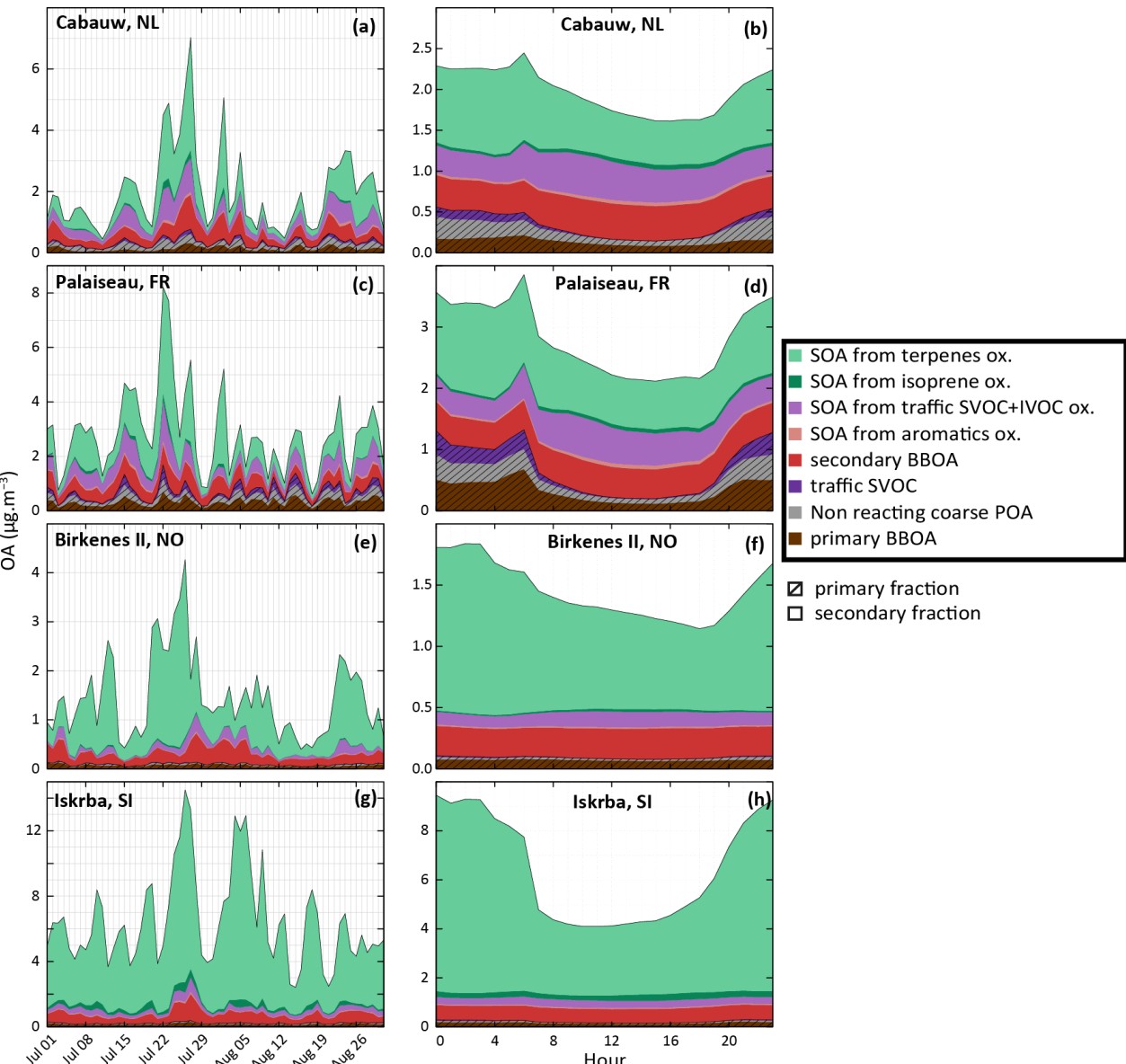

**Figure 14 – Evolution of simulated OA concentrations and distribution function of sources with the IVOC$_{30VOC}$ model configuration. Panels a, c, e and g present evolutions of daily average concentrations during the July-August 2013. Panels b, d, f and h present mean daily profiles. Results are shown at two stations influenced by anthropogenic sources in Netherland (Cabauw, a and b) and in France (Palaiseau, c and d) and at two stations influenced by biogenic sources in Norway (Birkenes II, e and f) and Slovenia (Iskrba, g and h). Primary and secondary BBOA includes compounds from biomass burning. Traffic SVOC includes C$_{14}$ to C$_{26}$ VBS-GECKO alkanes and alkenes and SOA from traffic SVOC+IVOC oxidation includes their oxidation products. SOA from terpenes includes all species produced by α-pinene, β-pinene, limonene, ocimene and humulene oxidation.**

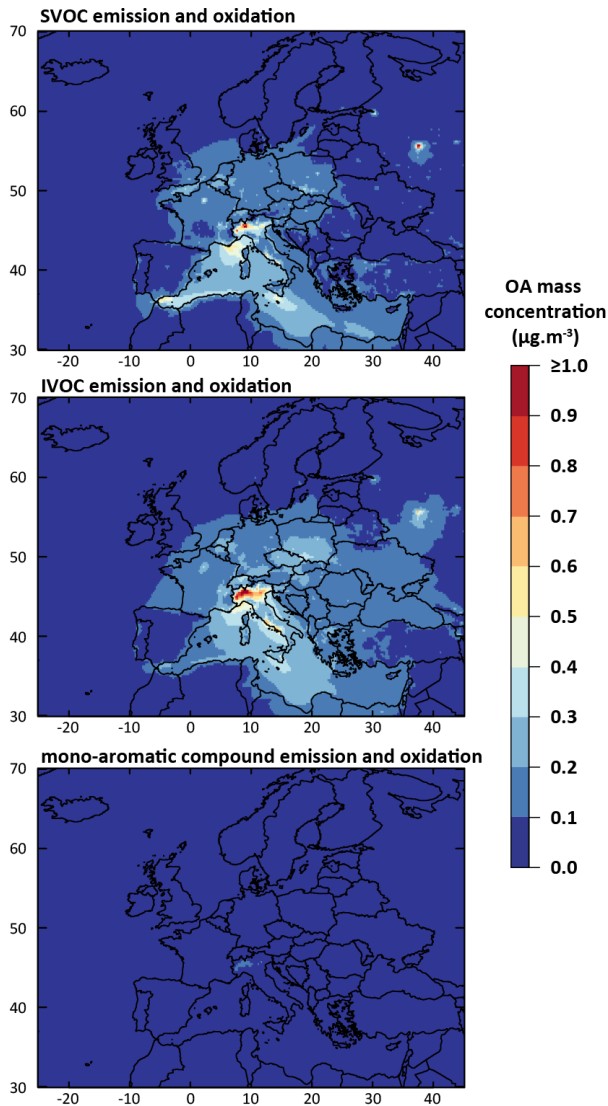

**Figure 15 – Mean simulated anthropogenic OA mass concentration formed by the partitioning of species produced from the oxidation of emitted traffic SVOCs (a), traffic IVOCs (b) and mono-aromatic compound (c) for July-August 2013 over Europe (data from IVOC$_{30VOC}$ simulation).**

