# Peer review of "Modeling organic aerosol over Europe in summer conditions with the VBS-GECKO parameterization: sensitivity to secondary organic compound properties and IVOC emissions"

_Atmospheric Chemistry and Physics, 2018_

## Referee Comment (RC1) · Anonymous Referee #1 · 29 Mar 2019

The manuscript presents a detailed analysis of the performances of VBS-GECKO SOA model implemented in the chemical transport model CHIMERE. The study is carefully done: the methodology and the results are well described; the simulated concentrations of organic carbon are compared to the observations and to other simulations in a consistent way. The period investigated is relevant for the purpose of the study and the input data already tested in a previous modelling exercise.

The results of the study are definitely relevant for the scientific community but still it will be interesting to discuss what happen when model configuration considers simul-

taneous the changes of two or more parametrization properties and also combinations between the IVOC emissions from vehicle exhausts and the parametrization properties.

Specific comments: Pg. 6, Line 19 I suggest to rephrase "ECMWF in the EURODELTA-III project was shown to be one of the most representative models over "Europe." in a meaningful way since ECMWF is not a model.

---

## Referee Comment (RC2) · Anonymous Referee #2 · 14 May 2019

Lannque et al. report on findings from simulations performed with a chemical transport model over Europe. They find that the VBS-GECKO model to simulate organic aerosol (OA) chemistry and gas/particle partitioning, which they argue is more physically-based, provides slight improvements in model performance. They also performed sensitivity simulations but, except for the inclusion of intermediate volatility organic compound (IVOC) emissions in urban areas, find the model results to exhibit low sensitivity to ranges explored in the input values.

This study adds to the growing literature on predicting the formation, composition, and

properties of secondary organic aerosol (SOA) on regional scales. SOA is an important, yet uncertain, component of PM2.5 and hence the study is well motivated. The VBS-GECKO model undertakes a novel approach to modeling OA where it leverages chemical detail from GECKO but in a heavily parameterized form. However, I have several concerns with the model detail provided in the manuscript and the interpretation of the simulation results (outlined below). I am hesitant to recommend publication in ACP at this point. I have made some major and minor comments for the authors to consider.

Major comments:

1. POA and SVOC: Throughout the manuscript, the use of the terms POA and SVOC is unclear. For example, on page 4, lines 21-24, the authors describe the SVOC/POA ratio used in this work. What does this mean? Are those the SVOCs in the gas-phase ratioed against the POA in the particle phase? If yes, how does the statement on line 21, 'POA from emission inventories are considered as SVOC' make sense? Also, is the ratio used to just determine the SVOC emissions and that the POA and SVOC are then repartitioned in the model? Also, a more elaborate justification of the source-resolved SVOC/POA ratios is warranted. A sentence saying 'This factor was shown to give satisfactory results' is insufficient.

2. POA: How is the POA dealt with in VBS-GECKO or is the POA formulation borrowed from H2O? If yes, I am a little concerned that the POA volatility distribution used in this work relies on that reported in Robinson et al. (2007). The Robinson parameterization is from a small off-road diesel engine and is probably not very representative of the gasoline- and diesel-powered sources in Europe. There has been significant amount of experimental (e.g., May et al., 2013a,b,c, Louvaris et al., 2017) and modeling (e.g., Woody et al., 2016, Jathar et al., 2017; Akherati et al., 2019) work since the Robinson paper in parameterizing and applying the POA volatility by source. While I would like to see these parameterizations be used in the base simulations, at the very least, insight from earlier work needs to be included as a sensitivity study to show the influence of source-resolved POA volatility on regional OA mass concentrations.

3. VBS-GECKO parameterization: The description for the OA model is incomplete (confusing at times) and needs to be significantly improved. Here are a few examples to help restructure this section: (i) How are IVOCs dealt with and what surrogates are used to model their SOA formation? (ii) How are IVOC emissions developed? (iii) Does this section describe the treatment of SVOCs too and if yes, what surrogates are used? I am concerned that the manuscript treats SVOCs and IVOCs interchangeably. (iv) How is the NOx dependence modeled? I am guessing that the ratios that are being talked about discuss the branching ratio between RO2 and NO and RO2 and HO2. Are there separate VBSs for products formed at different NOx levels? How are species formed at a high (low) NOx level then aged or transformed when they react further at a low (high) NOx level? All of these have implications on how results in the sensitivity section (4.3) are interpreted. (v) It is unclear whether each precursor gets a separate 7-bin VBS. Does it? If yes, the properties of the VBS bins remain the same but the mass yields are different? (vi) How is partitioning modeled for a given species between the organic and aqueous phases? Presumably, aqueous refers to the water associated with inorganic aerosol. Also, what about water uptake by the organic species itself? (vii) How are the saturation vapor pressures (Psat) for the VBS species determined? The Psat values are changed in the sensitivity simulations. A change in the Psat values by an order of magnitude should dramatically change the SOA produced. Wouldn't this affect the evaluation of GECKO predictions against chamber or flow reactor measurements? In fact, an evaluation of GECKO predictions was not discussed at all. Again, these have implications on how results in the sensitivity section (4.4) are interpreted. Finally, I think I understand very generally how GECKO is used to determine the mass yields and properties of the VBS product species but since this is the main contribution of this work, it would be better to provide a paragraph length description of how this is done and what those parameterizations represent.

4. Sensitivity simulations for ageing: I have major concerns in how the sensitivity simulations were performed and the interpretation drawn from the model predictions. Perhaps, some of these concerns stem from a lack of transparency in how the VBS-

GECKO parameterizations were developed and how ageing was modeled. For example, if the VBS-GECKO parameterizations are constrained to predictions from GECKO, which already includes ageing, why are another set of ageing reactions modeled in CHIMERE? In fact, how is ageing defined? Is it defined as the transformation past the time period, the GECKO simulations were run for? Also, if there is a need to model additional ageing, what is the justification for such a high kOH? Is this kOH value constrained? If yes, against what laboratory or field data? How is ageing modeled in VBS-GECKO? Does it use the 'bulldozer' mechanism of Robinson et al. (2007)? There is evidence that this type of ageing mechanism may overestimate SOA production in regional models (e.g., refer to Lane et al. (2008) for biogenic SOA and Jathar et al. (2016) for all SOA) and might require explicit modeling of fragmentation reactions (Shrivastava et al., 2013; 2015). Finally, the conclusion that 'aging rates are likely not a major source of uncertainty' needs to be framed in context of how it is modeled in this work and findings from earlier work.

5. IVOC simulations: I am not sure how varying the IVOC fraction of NMVOC represents a changing fleet. Presumably, the NMVOC emissions as part of the inventory already represent a fixed fraction of gasoline versus diesel sources. A changing fleet (e.g., more diesel and less gasoline) would reduce the total NMVOC emissions from mobile sources since diesels emit less NMVOC than gasoline per mile driven. So, a higher IVOC fraction could still result in a lower IVOC emission. Either this is part has not been described well or the emissions simulations are not accounting for changing total NMVOC emissions with a changing fleet. Also, it needs to be stated that these emissions are only being applied to mobile sources and also only to tailpipe emissions (i.e., not evaporative emissions). Is that right? 6. Grammar and style: There are numerous grammatical mistakes in the manuscript and the style could be improved too to enhance readability.

Minor comments:

1. Page 2, lines 5-9: Can this first paragraph include newer citations when discussing

OA and SOA?

2. Page 2, line 19: The SOA in chamber experiments rarely represents oxidation products after a single oxidation step. Rephrase.

3. Page 2, paragraph starting on line 16: This could benefit from reviewing the literature on a few other types of SOA modeling. For example, Jathar et al.'s (GMD, 2015) use of the statistical oxidation model. Or Li et al.'s (AE, 2015) use of the master chemical mechanism. In other words, there aren't just two approaches to model SOA but rather several methods that span a continuum in complexity.

4. Page 4, line 10: Is the particle size resolution too low to separate PM1 from PM2.5? Wouldn't numerical diffusion add uncertainty to how PM is split in that 1 to 2.5 $\mu$m region?

5. Page 4, line 22: The extensive emissions work done with open burning suggests that residential burning should have a lower SVOC/POA ratio than other anthropogenic combustion sources.

6. Section 2.2.2: To clarify, each precursor species in Table 1 gets a separate 7-bin VBS?

7. Page 5, line 16: Rephrase 'VBS-GECKO considers the formation of 7 volatility bins'.

8. Page 5, line 28: How is RRR calculated in CHIMERE?

9. Page 6, line 24: evaluated 'by' comparing.

10. Page 6, lines 24-32: Is there a reason for sub-selecting 7 stations for comparing the time series. What not do a comparison at all available sites? Can details about how the OA is measured (e.g., filter followed by OC/EC, AMS) added to this paragraph?

11. Page 7, line 14: Please comment on whether regulatory agencies use the Boylan and Russell (2006) criteria when evaluating model output. My sense is that they do not.

12. Page 7, lines 20-25: What is the primary reason for differences in the ref versus VBS-GECKO simulations for OA mass concentrations (e.g., source treatment, chemistry)? It is important to discuss this since the VBS-GECKO leads to slightly better model performance.

13. Page 7, line 30: Is this the Pearson r? If yes, please use the small letter 'r'? If not, please specific what correlation coefficient this is. Can you comment on the value of r? How does it compare to r from earlier regional modeling work (e.g., Ahmadov et al., JGR, 2012; Baker et al., ACP, 2015; Murphy et al., ACP, 2017)? It would be good to contrast the model skill across different geographical regions (i.e., United States versus Europe). To me an r2 of 0.25 doesn't sound that good. Does it?

14. Figure 3: It would improve interpretation of the data if the site location was specified in the panel instead of using the station number?

15. Page 9, line 8: I thought a VBS species could condense in both the organic and aqueous phases depending on the organic and aqueous partitioning coefficients respectively. If not, how is this treatment in the reference model handled? Perhaps, there is a need for a table that talks about the different model configurations?

16. Page 9, line 24: Where is Benelux? Is this location significant? If yes, please specify. Please also do so anywhere else results form a specific location are discussed.

17. Figure 5: General comment: Given the very small differences in panels (a) and (d), at what percent level would one be able to say with confidence that there were differences in model predictions? In other words, what is the precision in the model?

18. Section 4.1: The results presented in this section are sensitive to the amount of aerosol water available? How well is aerosol water modeled when compared to measurements? In fact, how well is inorganic aerosol modeled when compared to measurements?

19. Page 9, line 33: What does 'slightly' mean? Can you be quantitative?

20. Page 11, line 4: What does 'free NOx' mean?

21. Page 11, line 5: What does 'whatever' mean?

22. Page 12, line 21: I am not sure why IVOCs would ever be in the particle phase. In the hot exhaust, IVOCs are likely to be in the vapor phase. When hot exhaust is diluted and cooled, the IVOCs probably don't have enough time to condense and then evaporate.

23. Page 13, lines 1-13: Are the IVOCs described here referred to as SVOCs in the lines following this section (lines 14-31)? This was quite confusing.

24. Page 13, lines 14-31: I was hoping this would be covered in the methods section.

25. Figure 14: Could these also include the measurements?

26. Page 15, lines 15-21: Are newly discovered pathways for SOA from autooxidation reactions and acid-catalyzed pathways included? If not, could these be cited here to provide context?

27. Page 17, line 6: I do not recollect where the volatility of the OA was discussed and how the authors conclude that LVOC and ELVOC make up most of the OA.

---

## Author Comment (AC1) · 12 Nov 2019

**Author Responses to Referees and comments on**

**Modelling organic aerosol over Europe in summer conditions with the VBS-GECKO parameterization: sensitivity to secondary organic compound properties and IVOC emissions**

Victor Lannuque[1,2,3,a], Florian Couvidat[2], Marie Camredon[1], Bernard Aumont[1] and Bertrand Bessagnet[2,b]

[1] LISA, UMR CNRS 7583, IPSL, Université Paris Est Créteil and Université de Paris, 94010 Créteil Cedex, France.
[2] INERIS, National Institute for Industrial Environment and Risks, Parc Technologique ALATA, 60550 Verneuil-en-Halatte, France.
[3] Agence de l'Environnement et de la Maîtrise de l'Energie, 20 avenue du Grésillé - BP 90406, 49004 Angers Cedex 01, France.
[a] Now at: CEREA, Joint Laboratory École des Ponts ParisTech – EDF R & D, Université Paris-Est, 77455 Marne la Vallée, France.
[b] Now at: Laboratoire de Météorologie Dynamique, IPSL, CNRS, UMR8539, 91128 Palaiseau Cedex, France.

*Correspondence to*: Victor Lannuque (victor.lannuque@lisa.u-pec.fr.com) and Florian Couvidat (florian.couvidat@ineris.fr)

We thank the reviewers for their comments on the manuscript. We outline below responses to the points raised by each referee and summarize the changes made to the revised manuscript.

**Reponses to Referee 1**

> *The results of the study are definitely relevant for the scientific community but still it will be interesting to discuss what happen when model configuration considers simultaneous the changes of two or more parameterization properties and also combinations between the IVOC emissions from vehicle exhausts and the parameterization properties.*

Sensitivity tests were run to (1) test the robustness of the VBS-GECKO parameterization to the uncertainties of the VBS-GECKO species properties and (2) quantify the sensitivity of the model results to IVOC emissions. Due to the high computation cost of these simulations, it was necessary to limit the number of sensitivity tests. The

objectives (1) and (2) are dissociated here and the sensitivity tests on each VBS-GECKO parameterization property and IVOC emission configuration were therefore run separately. Results on objective (1) show that the uncertainties in some parameter values of the VBS parameterization have a limited impact on the formation of SOA, and that therefore the parameterization is robust enough for 3D applications. Running tests simultaneously would be of important calculation costs and would not bring additional information on the robustness of the parameterization.

> *Specific comments: Pg. 6, Line 19 I suggest to rephrase "ECMWF in the EURODELTA-III project was shown to be one of the most representative models over Europe." in a meaningful way since ECMWF is not a model.*

You are right. Here "ECMWF" is related to IFS (Integrated Forecasting System) model from ECMWF. "ECMWF" has been replaced by "ECMWF-IFS" in the manuscript.

**Reponses to Referee 2**

> *1. POA and SVOC: Throughout the manuscript, the use of the terms POA and SVOC is unclear. For example, on page 4, lines 21-24, the authors describe the SVOC/POA ratio used in this work. What does this mean? Are those the SVOCs in the gas-phase ratioed against the POA in the particle phase? If yes, how does the statement on line21, 'POA from emission inventories are considered as SVOC' make sense? Also, is the ratio used to just determine the SVOC emissions and that the POA and SVOC are then repartitioned in the model? Also, a more elaborate justification of the source-resolved SVOC/POA ratios is warranted. A sentence saying 'This factor was shown to give satisfactory results' is insufficient.*

In the manuscript, "POA" is always related to the primary organic aerosol provided in the emission inventory. This POA, originally considered as non-volatile, has been shown since the study by Robinson et al. (2007) to be a set of semi-volatile species. Here, "SVOC(s)" corresponds to this set of semi-volatile species. In this study, all emitted POA were treated as SVOCs according the Robinson et al. (2007) volatility distribution and fraction (SVOC/POA ratio equal to 1, see Figure 11). Depending on their volatility, these SVOCs partition in the model between the gas and the particle.

The term "SVOC/POA ratio" is indeed not clear in the manuscript, as it has been used to modify the emission fluxes. The term "SVOC/POA ratio" was removed in the revised version of the manuscript and the sentences "a SVOC/POA ratio of 5 is used assuming that wood burning emissions are underestimated (e.g. Denier Van Der Gon et al., 2015). This factor was shown to give satisfactory results on OA estimations (Couvidat et al. 2012, 2018)." were changed into "POA emitted from residential wood burning have been shown to be significantly underestimated over Europe (e.g. Denier Van Der Gon et al., 2015). In this study, POA emissions from residential sources have therefore been increased by a factor of 5 as advised by Couvidat et al. (2012, 2018)."

*> 2. POA: How is the POA dealt with in VBS-GECKO or is the POA formulation borrowed from H2O? If yes, I am a little concerned that the POA volatility distribution used in this work relies on that reported in Robinson et al. (2007). The Robinson parameterization is from a small off-road diesel engine and is probably not very representative of the gasoline- and diesel-powered sources in Europe. There has been significant amount of experimental (e.g., May et al., 2013a,b,c, Louvaris et al., 2017) and modeling (e.g.,Woody et al., 2016, Jathar et al., 2017; Akherati et al., 2019) work since the Robinson paper in parameterizing and applying the POA volatility by source. While I would like to see these parameterizations be used in the base simulations, at the very least, insight from earlier work needs to be included as a sensitivity study to show the influence of source-resolved POA volatility on regional OA mass concentrations.*

Significant amount of experimental and modeling studies has indeed been published on SVOC and IVOC emissions since the Robinson et al. (2007) paper. The reference to these studies are now included in the revised version of the manuscript.

The purpose of this paper is the implementation and the evaluation of the VBS-GECKO parameterization in a CTM. The reference simulations were performed with the latest published version of $H^2O$ (Couvidat et al., 2018) that uses the SVOC distribution of POA emissions from Robinson et al. (2007). This POA treatment was thus kept unchanged in this study. Possible errors in the simulated OA mass can indeed be due to the use of the Robinson et al. (2007) to represent the volatility distribution of POA emitted by all activity sectors. A discussion on this has been added in the conclusion of the manuscript.

For the evaluation of the VBS-GECKO parameterization (sections 3 and 4), the emitted POA distribution into S/IVOC volatility bins, as well as the gaseous oxidation chemical mechanisms for these S/IVOCs, are thus the same than in the latest published version of the $H^2O$ (Couvidat et al., 2018), used here as a reference. For the study of OA sensitivity on IVOC emissions using the VBS-GECKO parameterization (section 5), the POA

distribution into S/IVOC volatility bins is different for traffic sources (SNAP 7 and 8). This new distribution of POA emissions still follows the volatility distribution provided by Robinson et al. (2007), but using the VBS-GECKO species to fill 9 volatility bins (i.e. linear n-alkane and 1-alkene species (C14 to C26) as shown in Figure 11), and the VBS-GECKO parameterization to represent the gaseous oxidation chemical mechanism of these S/IVOCs.

> *3. VBS-GECKO parameterization: The description for the OA model is incomplete (confusing at times) and needs to be significantly improved.*

Most of these questionings deals with the architecture, the optimization and the evaluation of the VBS-GECKO parameterization. These points are described in detail in a previous paper (Lannuque et al., 2018). It appears however from this comment that not enough information on the VBS-GECKO parameterization is provided in this manuscript. The section describing the VBS-GECKO parameterization was therefore modified to include additional information on its structure and its relationship with the GECKO-A explicit mechanisms (section 2.2.2). Specific answers to the referee comments are provided below.

> *Here are a few examples to help restructure this section:*
*(i) How are IVOCs dealt with and what surrogates are used to model their SOA formation?*
*(ii) How are IVOC emissions developed?*
*(iii) Does this section describe the treatment of SVOCs too and if yes, what surrogates are used? I am concerned that the manuscript treats SVOCs and IVOCs interchangeably.*

The VBS-GECKO parameterization is a VBS type parameterization that represents the formation of SOA from a set of defined primary species: aromatic compounds (benzene, toluene, and o- m- and p-xylenes), biogenic compounds (α-pinene, β-pinene and limonene), linear n-alkanes (tetradecane, octadecane, docosane, hexacosane) and linear 1-alkenes (tetradecene, octadecene, docosene, hexacosene).

In this study, SOA formation from NMVOC emissions was taken into account for mono-aromatic compounds, terpenes and long carbon chain NMVOCs. These long carbon chain NMVOCs have a carbon number lower or equal to 13 (organic compounds having a carbon number greater or equal to 14 being not included in NMVOC emission inventories as their vapor pressure fall in the IVOC range). α-pinene, β-pinene and limonene VBS-

GECKO precursors were used as surrogate species to represent the terpenes. Benzene, toluene, and o- m- and p-xylene VBS-GECKO precursors were used to represent the mono-aromatic compounds, and the C10 and C14 linear n-alkanes and 1-alkenes to represent the long carbon chain NMVOCs.

SVOC emission factors were taken into account considering the volatility distribution provided by Robinson et al. (2007). In section 3 and 4, this volatility distribution was filled up using the 3 precursors present in the $H_2O$ parameterization, having saturation vapor pressures at 298 K of $8.9 \times 10^{-11}$, $8.4 \times 10^{-9}$ and $3.2 \times 10^{-7}$ atm respectively In sections 5, the SVOC volatility distribution was filled up using the VBS-GECKO precursors, i.e. with a mixing of C18, C22 and C26 linear n-alkanes and 1-alkenes (see Figure 11).

IVOC emissions were not considered in section 3 and 4. In section 5, IVOC emission factors were taken into account considering the volatility distribution of inferred IVOCs provided by Robinson et al. (2007). the IVOC volatility distribution was filled up using the VBS-GECKO precursors, i.e. with a mixing of C14 and C18 linear n-alkanes and 1-alkenes (see Figure 11).

*(iv) How is the NOx dependence modeled? I am guessing that the ratios that are being talked about discuss the branching ratio between RO2 and NO and RO2 and HO2. Are there separate VBSs for products formed at different NOx levels? How are species formed at a high (low) NOx level then aged or transformed when they react further at a low (high) NOx level? All of these have implications on how results in the sensitivity section (4.3) are interpreted.*

The stoichiometric coefficients of the VBS-GECKO parameterization (called $a_{k,RRR,i}$, where k is the precursor and i the number of the volatility bin) are NOx dependent. This NOx dependence is represented according to fraction of $RO_2$ reacting with NO, called the RRR ratio, calculated as:

$$RRR = \frac{k_{RO_2+NO}[NO]}{k_{RO_2+NO}[NO]+k_{RO_2+HO_2}[HO_2]}$$

where $k_{RO2+NO}$ and $k_{RO2+HO2}$ are the rate constants for the reactions of the peroxy radicals with NO (set to $9.0 \times 10^{-12}$ $cm^3$ $molec^{-1}$ $s^{-1}$ according Jenkin et al., 1997 at 298 K) and with $HO_2$ (set to $2.2 \times 10^{-11}$ $cm^3$ $molec^{-1}$ $s^{-1}$ according to Boyd et al., 2003, assuming a large carbon skeleton for $RO_2$ at 298 K), respectively. Each stoichiometric coefficient has 5 tabulated values corresponding to RRRs of 0, 0.1, 0.5, 0.9 and 1. The entire range of NOx conditions is covered by a linear interpolation of the coefficients between the two tabulated RRR values

that encompass the calculated RRR. The formed $VB_{k,i}$ for the precursor k and the volatility bin i are not dependent on level of NOx. More details on the NOx dependence of the stoichiometric coefficients were added to the revised version.

*(v) It is unclear whether each precursor gets a separate 7-bin VBS. Does it? If yes, the properties of the VBS bins remain the same but the mass yields are different?*

Yes. In the VBS-GECKO parameterization, each precursor k has its own set of (1) 7 volatility bins, called $VB_{k,i}$ where i is the bin number, (2) 35 stoichiometric coefficients $a_{k,RRR,i}$, for the formation of the $VB_{k,i}$ during the gaseous reaction of the precursor with OH, plus 35 stoichiometric coefficients $b_{k,RRR,i}$ if the precursor k reacts also with $O_3$, and 35 stoichiometric coefficients $c_{k,RRR,i}$ if the precursor k reacts with $NO_3$, (3) 210 stoichiometric coefficients $d_{k,RRR,i,j}$ for the formation of the $VB_{k,j}$ during the gaseous reaction with OH of the 6 more volatile $VB_{k,i}$ and (4) 1 factor $\phi_k$ for the photolysis in the gas phase of the 6 more volatile $VB_{k,i}$. The properties of each $VB_{k,i}$ (i.e. molar weights (Mw), saturation vapor pressures ($P^{sat}$) at 298 K, effective Henry's law constants ($H^{eff}$) at 298K and vaporizations enthalpies ($\Delta H_{vap}$)) are the same for all precursor. The structure of the VBS-GECKO parameterization was added to the manuscript.

*(vi) How is partitioning modeled for a given species between the organic and aqueous phases? Presumably, aqueous refers to the water associated with inorganic aerosol. Also, what about water uptake by the organic species itself?*

The gas/aerosol partitioning of the precursors and VBk,i from the VBS-GECKO was taken into account for species having a low volatility, using the Raoult's law and considering an equilibrium between the gas phase and an organic aerosol phase. In the section dealing with the sensitivity tests on the hydro-solubility of organic compounds (section 4.1), the gas/aerosol partitioning was taken into account for species having a low volatility and also a high solubility, using the Raoult's and the Henry's law respectively, and considering the equilibrium between the gas phase, an organic aerosol phase and an aqueous aerosol phase. The gas/aerosol partitioning was calculated in this study using the SOAP thermodynamic module (Couvidat and Sartelet, 2015). These details were added to the revised version.

The influence of organic species on the aerosol hygroscopicity was not taken into account in this study.

*(vii) How are the saturation vapor pressures (Psat) for the VBS species determined? The Psat values are changed in the sensitivity simulations. A change in the Psat values by an order of magnitude should dramatically change the SOA produced. Wouldn't this affect the evaluation of GECKO predictions against chamber or flow reactor measurements? In fact, an evaluation of GECKO predictions was not discussed at all. Again, these have implications on how results in the sensitivity section (4.4) are interpreted. Finally, I think I understand very generally how GECKO is used to determine the mass yields and properties of the VBS product species but since this is the main contribution of this work, it would be better to provide a paragraph length description of how this is done and what those parameterizations represent.*

The GECKO-A chemical mechanisms were evaluated in several publications by comparisons with chamber experiments (e.g. Valorso et al., 2011; Denjean et al., 2015; La et al., 2016; McVay et al., 2016) and field campaign measurements (e.g. Lee-Taylor et al., 2011, 2015; Hodzic et al., 2013). The evaluation of GECKO-A chemical mechanisms is largely out of the scope of this study.

The development and the evaluation of the VBS-GECKO parameterization on the basis of GECKO-A explicit chemical mechanisms was the purpose of the Lannuque et al. (2018) paper. This paper describes in particular (1) the optimization of the VBS-GECKO parameters on a learning set of GECKO-A explicit simulations and (2) the evaluation of the VBS-GECKO parameterization by comparison with a validation set of GECKO-A explicit simulations. As mentioned above, the section describing the VBS-GECKO parameterization was modified in order to improve the understanding.

In this work, sensitivity tests were performed on the VBS-GECKO parameterization, and not on the GECKO-A explicit chemical mechanisms. In the GECKO-A tool, the physico-chemical properties of organic species (such as Psat298K) are estimated using structure-activity relationships (SAR). Comparisons between different SARs have shown that estimated Psat can vary by more than one order of magnitude (Valorso et al., 2011). The mandatory use of SAR in GECKO-A necessarily induces uncertainties, which affects the VBS-GECKO parameterization. The sensitivity tests on Psat are therefore performed to estimate the impact of this uncertainties on OA mass.

In the VBS-GECKO parameterization, the saturation vapor pressures ($P^{sat}$) of the VBk,i are fixed values at 298 K (see Table 1). The number of volatility bins and the values of the $P^{sat}$ of each bin were determined during the development of the parameterization and are presented in Lannuque et al. (2018). I quote Lannuque et al. (2018): "Tests were performed to establish the best number and range of volatility bins to get a compromise between the reliability and the size of the VBS-GECKO parameterization. The partitioning of organic species having a Psat298K lower than $10^{-12}$ atm is almost exclusively in the particulate phase under typical atmospheric conditions. Therefore, the lower volatility bin in the VBS-GECKO parameterization lumps all species having a $P^{sat}_{298K}$ lower than $10^{-12.5}$ atm. Similarly, organic species having a $P^{sat}_{298K}$ above $10^{-6}$ atm are dominantly in the gas phase under typical atmospheric conditions and volatility bin is included for species having a $P^{sat}_{298K}$ above $10^{-5.5}$ atm. Between these thresholds, the gas/particle partitioning of a species depends sensitively on temperature and OA load, so that a finer volatility discretization is desirable. A total of 7 volatility bins were selected for the VBS-GECKO parameterization. The boundaries of the k volatility bins i, $VB_{k,i}$, are the same for all the precursors k. Bin intervals at 298 K are shown in Table 2 and in Figure 5 (of Lannuque et al., 2018) by the different colours of the SOC bubbles. The properties of the k volatility bins i, VBk,i, are set to the same value for the various precursors k. Each of the VBk,i has for assigned saturation vapour pressure the central value of its logarithmic interval".

> *4. Sensitivity simulations for ageing: I have major concerns in how the sensitivity simulations were performed and the interpretation drawn from the model predictions. Perhaps, some of these concerns stem from a lack of transparency in how the VBS-GECKO parameterizations were developed and how ageing was modeled. For example, if the VBS-GECKO parameterizations are constrained to predictions from GECKO, which already includes ageing, why are another set of ageing reactions modeled in CHIMERE? In fact, how is ageing defined? Is it defined as the transformation past the time period, the GECKO simulations were run for?*

In the VBS-type parameterizations, the term "aging" is generally associated with all the reactions impacting secondary organic species, and leading to functionalization, fragmentation and/or oligomerization of the organic matter. Here, the term "aging" is therefore associated with the "$VB_{k,i}$ + OH" and "$VB_{k,i}$ + hv" gaseous reactions, for i equal 1 to 6. This aging is indeed implicitly included in GECKO-A chemical mechanisms, and is considered in the architecture of the VBS-GECKO parameterization with stoichiometric coefficients for the formed $VB_{k,j}$, (for j equal 1 to 7) optimized on the basis of GECKO-A simulations (Lannuque et al., 2018). No additional set of aging reactions were considered in the VBS-GECKO-A parameterization for its use in CHIMERE. A sentence

was added to the manuscript to define the term "aging". The description of the VBS-GECKO parameterization structure in the revised version will also help in clarifying how the aging is represented.

> *Also, if there is a need to model additional ageing, what is the justification for such a high kOH? Is this kOH value constrained? If yes, against what laboratory or field data?*

In the VBS-GECKO parameterization, a $k_{OH}$ of $4.10^{-11}$ $cm^3$ $molec^{-1}$ $s^{-1}$ is used for all the $VB_{k,i}$, for i equal 1 to 6. This value is in the upper range of the $k_{OH}$ values estimated for the secondary organic compounds explicitly simulated with GECKO-A (see Lannuque et al., 2018). In the building of the VBS-GECKO parameterization, this $k_{OH}$ value is first fixed, and then stoichiometric coefficients are optimized. A high value of $k_{OH}$ for the reactivity of a $VB_{k,i}$ + OH is thus compensated by lower values of stoichiometric coefficients related to the formation of the $VB_{k,j}$ (with j different from i) and a higher value of the coefficient related to the reformation of the $VB_{k,j}$. In the manuscript, sensitivity tests were performed on this $k_{OH}$ value to indirectly evaluate the stability of the VBS-GECKO parameterization to the uncertainty in the simulated OH concentrations.

> *How is ageing modeled in VBS-GECKO? Does it use the 'bulldozer' mechanism of Robinson et al. (2007)? There is evidence that this type of ageing mechanism may overestimate SOA production in regional models (e.g., refer to Lane et al. (2008) for biogenic SOA and Jathar et al. (2016) for all SOA) and might require explicit modeling of fragmentation reactions (Shrivastava et al., 2013; 2015).*

For each precursor k, the gaseous fraction of the $VB_{k,i}$ (for i equal 1 to 6) reacts with OH to form the corresponding 7 $VB_{k,j}$ (for j equal 1 to 6) (see Table S2 and Lannuque et al., 2018). By definition, the volatility of the $VB_{k,i}$ decreases as i increases. Thus in the VBS-GECKO parameterization, the reaction of the $VB_{k,i}$ with OH considers both fragmentation and functionalization pathways, the formation of a $VB_{k,j}$ with j < i being associated with fragmentation and the formation of a $VB_{k,j}$ with j > i being associated with functionalization. Unlike the mechanism of Robinson et al. (2007), fragmentation is therefore taken into account. To improve the manuscript clarity, the terms "fragmentation" and "functionalization" are now mentioned in the description of VBS-GECKO.

>*Finally, the conclusion that 'aging rates are likely not a major source of uncertainty' needs to be framed in context of how it is modeled in this work and findings from earlier work.*

You are right. This conclusion is related to the sensitivity tests performed in this study on the $k_{OH}$ value of the VBS-GECKO papameterization. It is now mentioned in the text.

> *5. IVOC simulations: I am not sure how varying the IVOC fraction of NMVOC represents a changing fleet. Presumably, the NMVOC emissions as part of the inventory already represent a fixed fraction of gasoline versus diesel sources. A changing fleet (e.g., more diesel and less gasoline) would reduce the total NMVOC emissions from mobile sources since diesels emit less NMVOC than gasoline per mile driven. So, a higher IVOC fraction could still result in a lower IVOC emission. Either this is part has not been described well or the emissions simulations are not accounting for changing total NMVOC emissions with a changing fleet.*

You are right. Sensitivity tests were performed on total IVOC emissions. IVOC emissions were estimated as 150% of POA emissions (Robinson et al., 2007), and 4, 16, 30 or 65% of NMVOC emissions (Zhao et al., 2015, 2016). The term "fleet change" is thus not the right term. This was modified in the abstract, in the introduction, in section 5 and in the conclusion.

> *Also, it needs to be stated that these emissions are only being applied to mobile sources and also only to tailpipe emissions (i.e., not evaporative emissions). Is that right?*

The sensitivity tests on IVOC emissions were performed on all "road transport" emissions (SNAP 7) and all "other mobile source and machinery" emissions (SNAP 8). SNAP 7 includes tailpipe emissions but also "evaporation of gasoline from vehicles" which are thus both considered in our study (SNAP 7 also includes emissions due to abrasion of tires and brake pads and to road wear which are negligible in term of organic compound emissions). These precisions were added to the revised version.

> *6. Grammar and style: There are numerous grammatical mistakes in the manuscript and the style could be improved too to enhance readability.*

The manuscript has been re-read and corrected to try to enhance readability.

> *Minor comments:*
> *1. Page 2, lines 5-9: Can this first paragraph include newer citations when discussing OA and SOA?*

Recent citations were added to the introduction of the revised version of the manuscript.

> *2. Page 2, line 19: The SOA in chamber experiments rarely represents oxidation products after a single oxidation step. Rephrase.*

The sentence was not about SOA in chamber experiments but about the Odum approach. This sentence was anyway removed from the manuscript as this paragraph was modified to take into account the next comment.

> *3. Page 2, paragraph starting on line 16: This could benefit from reviewing the literature on a few other types of SOA modeling. For example, Jathar et al.'s (GMD, 2015) use of the statistical oxidation model. Or Li et al.'s (AE, 2015) use of the master chemical mechanism. In other words, there aren't just two approaches to model SOA but rather several methods that span a continuum in complexity.*

As suggested by the referee, the paragraph was modified to list the various types of SOA parameterizations for 3D applications. The use of the MCM chemical scheme in CTMs was not added here as this approach is still rarely applied because of important calculation costs.

> *4. Page 4, line 10: Is the particle size resolution too low to separate PM1 from PM2.5? Wouldn't numerical diffusion add uncertainty to how PM is split in that 1 to 2.5µm region?*

The size resolution of particles in the model allows thus to separate $PM_1$ and $PM_{2.5}$ (9 size bins from 10 nm to 10 µm with especially fixed cut off at 1 µm and 2.5 µm). The particular matter is distributed into these bins depending to its origin (biogenic, anthropogenic, dust, etc). The two sizes are used in the study as the available in-situ data are OM in $PM_1$ (ACSM) and OC in $PM_{2.5}$.

> *5. Page 4, line 22: The extensive emissions work done with open burning suggests that residential burning should have a lower SVOC/POA ratio than other anthropogenic combustion sources.*

The term "SVOC/POA ratio" was unclear and removed in the revised version of the manuscript (see reply to general comment #1).

> *6. Section 2.2.2: To clarify, each precursor species in Table 1 gets a separate 7-binVBS?*

Exactly. For a better understanding, the volatility bins are now called $VB_{k,i}$, where k is the precursor and i the bin number (see reply to general comment #3.v).

> *7. Page 5, line 16: Rephrase 'VBS-GECKO considers the formation of 7 volatility bins'.*

It was replaced by: "In the VBS-GECKO parameterization, secondary organic species formed during the oxidation of a given precursor k are lumped into 7 volatility bins."

> *8. Page 5, line 28: How is RRR calculated in CHIMERE?*

RRR is the fraction of $RO_2$ reacting with NO (assuming that $RO_2+RO_2$ can be considered as negligible as done by Lane et al., 2008). RRR is calculated in each box at each chemical time step following the formula provided in the answer to the general comment #3.iv). This information was added to the manuscript.

> *9. Page 6, line 24: evaluated 'by' comparing.*

The text was modified.

> *10. Page 6, lines 24-32: Is there a reason for sub-selecting 7 stations for comparing the time series. What not do comparison at all available sites? Can details about how the OA is measured (e.g., filter followed by OC/EC, AMS) added to this paragraph?*

A total of 48 rural background stations was available for the comparison. The evaluation of the model was shown in the article using statistical indicators calculated on these 48 rural stations. Most of these stations provide $PM_{2.5}$ measurement and/or only few measurement points on the studied temporal period. All the stations were thus not of interest for temporal evolution comparisons of the aerosol organic fraction. Only stations providing $OM_{PM1}$ and $OC_{PM2.5}$ measurements with a large amount of data on the studied period were selected. Measurements of OC

are performed by calcinations of filters and $OM_{PM1}$ using an ACSM. This information was added to the manuscript.

> *11. Page 7, line 14: Please comment on whether regulatory agencies use the Boylan and Russell (2006)*
> *criteria when evaluating model output. My sense is that they do not.*

Regulatory agencies do not use the Boylan and Russell critera. It is still a criterion which has been used in different studies to evaluate the model performances (e.g. Couvidat et al., 2018; Lecœur and Seigneur, 2013; Jiang et al., 2019 ACPD; Ciarelli et al., 2017; Mircea et al., 2019). The references were added to the revised version.

> *12. Page 7, lines 20-25: What is the primary reason for differences in the ref versus VBS-GECKO simulations*
> *for OA mass concentrations (e.g., source treatment, chemistry)? It is important to discuss this since the VBS-*
> *GECKO leads to slightly better model performance.*

The purpose of the comparison between the $H^2O$ and GECKO-VBS mechanism for SOA formation is to evaluate the reliability of the VBS-GECKO parameterization. The simulations performed with CHIMERE were therefore set up using the same configuration of the model (meteorological data, emissions, deposition, inorganic and organic gaseous chemical mechanism, inorganic and organic gas/particle partitioning…), but implementing either the $H^2O$ or the GECKO-VBS parameterization for SOA formation. The implementation a given parameterization for SOA formation induces anyway some differences depending on the primary compounds considered and/or the processes taken into account in the parameterization. Here, the differences between the $H^2O$ and VBS-GECKO in reference simulations come from the representation of:

(1) the secondary organic compounds that partition to form SOA. The $H^2O$ parameterization is built to represent SOA formation from the condensation of hydrophobic species and hydrophilic species. In the reference $H^2O$ simulation, the hydrophobic species were considered to condense only on an organic phase (depending on their saturation vapor pressure) and the hydrophilic species considered to condense both, on an organic phase and on an aqueous phase (depending on their Henry's constant). The VBS-GECKO parameterization is built to represent SOA formation from the condensation of species having a low volatility. In the simulation performed with the VBS-GECKO parameterization, the species were considered to condense on an organic phase only (depending on their saturation vapor pressure). The

differences between the two representations come from the consideration of SOA formation from the condensation of species with a high hydro-solubility in the $H_2O$ parameterization, processes which is not included in the VBS-GECKO parameterization. Even if in the sensitivity test on the hydro-solubility (Section 4.1), the species were considered to condense both on an organic phase and an aqueous phase, the VBS-GECKO was not developed for this purpose (see response to minor comments #15).

(2) The ideality of the condensed phase. In the $H^2O$ reference (i.e. the one evaluated by Couvidat et al., 2018), the organic and aqueous phases of the particle are considered as non-ideal, with estimations of activity coefficients. In the VBS-GECKO simulations, the particles were considered as ideal because of the difficulty to provide any estimation for activity coefficients.

(3) the SOA formation from emitted $C_{10}$ to $C_{13}$ precursors (from NMVOC inventory). These SOA precursors, included in the VBS-GECKO parameterization but not in the $H^2O$ parameterization, are only considered when simulations are performed with the VBS-GECKO parameterization. The contribution of $C_{10}$ to $C_{13}$ precursors to the OA mass has however been shown to be very low and do not explain the differences between the $H_2O$ and VBS-GECKO results.

The difference between the $H^2O$ and VBS-GECKO results, and the slightly better model performances calculated with the VBS-GECKO parameterization, are thus expected to come dominantly from the parameterization used, as well as point (1) and (2) to represent SOA formation. A paragraph was added in section 2.2 of the revised version to better introduce the differences between the $H^2O$ and VBS-GECKO reference simulations.

> *13. Page 7, line 30: Is this the Pearson r? If yes, please use the small letter 'r'? If not, please specific what correlation coefficient this is. Can you comment on the value of r? How does it compare to r from earlier regional modeling work (e.g., Ahmadov et al., JGR, 2012; Baker et al., ACP, 2015; Murphy et al., ACP, 2017)? It would be good to contrast the model skill across different geographical regions (i.e., United States versus Europe). To me an r2 of 0.25 doesn't sound that good. Does it?*

This is the Pearson r. "R" was replaced by "r" in the text.
For CTM/measurement comparisons, standard r values are low notably because of the large number of data, variations and phenomena to be taken into account as meteorology and air mass transportation. Our r values (0.79 for OM and 0.58 for OC) are in the standard. For example, in the article by Akherati et al. (2019) that you quote, the 6 calculated r values on OM are between 0.11 and 0.56. The other quoted references give r values between 0.55 and 0.7 for OM (Murphy et al., 2017) or between 0.55 and 0.76 for OC (Ahmadhov et al., 2012) in USA.

Bergström et al. (2012) found r values around 0.6 for OM at a station in Switzerland. Ciarelli et al. (2017) calculated r values at different stations in Europe and highlight that the lower correlations are found in mountainous area, suggesting a likely important impact of modeled dynamics of air masses on this statistical value, more than the chemistry. Text was modified and references added.

> *14. Figure 3: It would improve interpretation of the data if the site location was specified in the panel instead of using the station number?*

It was modified in the text and the Figures.

> *15. Page 9, line 8: I thought a VBS species could condense in both the organic and aqueous phases depending on the organic and aqueous partitioning coefficients respectively. If not, how is this treatment in the reference model handled? Perhaps, there is a need for a table that talks about the different model configurations?*

The VBS-GECKO parameterization, as most of the one-dimension VBS parameterization types, was built to represents SOA formation from the partitioning onto an organic aerosol phase of organic compounds having a low volatility. A Henry's law constant was added to each volatility bin species to calculate their deposition. This parameter can also be used to represent their condensation onto an aqueous phase. However, the VBS-GECKO parameterization (1) does not include the major precursors of SOA formation from the condensation of hydro-soluble species and (2) was not optimised to represent this process. The VBS-GECKO reference simulation considers thus SOA formation from the condensation of organic species having a low volatility only. A test was performed Section 4.1 to evaluate the sensitivity of the model to the condensation of these organic species on an aqueous phase depending on their Henry's law constant. The text was modified to clarify this point.

> *16. Page 9, line 24: Where is Benelux? Is this location significant? If yes, please specify. Please also do so anywhere else results form a specific location are discussed.*

Benelux means "Belgium-Netherland-Luxembourg". It is now clarified in the text.

> *17. Figure 5: General comment: Given the very small differences in panels (a) and (d), at what percent level would one be able to say with confidence that there were differences in model predictions? In other words, what is the precision in the model?*

All the simulations were carried out with the same inputs and on the same numerical environment. Moreover, CHIMERE use numbers in double precision. Simulation tests are regularly performed with CHIMERE, were also performed here. The differences observed here are because of the use of the different parameterizations.

> *18. Section 4.1: The results presented in this section are sensitive to the amount of aerosol water available? How well is aerosol water modeled when compared to measurements? In fact, how well is inorganic aerosol modeled when compared to measurements?*

Yes, the results presented Section 4.1 on the OA sensitivity to the hydro-solubility of organic species are indeed influenced by the amount of water available in the aerosol. However, in the model configuration, the condensation of the VBS-GECKO species on the aqueous phase does not impact the inorganic nor the water content of aerosol. The comparison is therefore coherent.

Water and inorganic composition of aerosols are calculated with ISORROPIA which is widely used in air quality models. The evaluation of inorganics in CHIMERE for the simulated period was studied in details in Couvidat et al. (2018). It was not possible to evaluate the amount of water during this period due to the absence of measurements. However, Guo et al. (2015) estimated over southeastern USA that amounts of water calculated with ISORROPIA were in good agreement with measured liquid water based on differences in ambient and dry light scattering coefficients and Bougiatioti et al. (2016) show similar results over eastern Mediterranean.

> *19. Page 9, line 33: What does 'slightly' mean? Can you be quantitative?*

Statistical values were incorporated into the text to be more quantitative.

> *20. Page 11, line 4: What does 'free NOx' mean?*

"free NOx" was replaced by "remote NOx conditions" in the manuscript.

> *21. Page 11, line 5: What does 'whatever' mean?*

The entire range of RRR ratio (from remote to high NOx conditions) is covered over Europe in every model configuration. Text was modified.

> *22. Page 12, line 21: I am not sure why IVOCs would ever be in the particle phase. In the hot exhaust, IVOCs are likely to be in the vapor phase. When hot exhaust is diluted and cooled, the IVOCs probably don't have enough time to condense and then evaporate.*

You are right, it is a mistake. The sentence was removed from the text.

> *23. Page 13, lines 1-13: Are the IVOCs described here referred to as SVOCs in the lines following this section (lines 14-31)? This was quite confusing.*

The text was modified to be more understandable and to be clear on what treatments are apply to SVOCs (distribution in low volatility bins from Robinson et al. (2007) corresponding to C18, C22 and C26 in VBS-GECKO parameterization), to IVOCs (relative distribution in intermediate volatility bins from Robinson et al. (2007) corresponding to C14 and C18 in VBS-GECKO parameterization, and absolute flux of emission depending on model configuration) or both S/IVOCs (considered as 25% alkenes and 75% alkanes). **IVOC$_{150POA}$**, **IVOC$_{4VOC}$**, **IVOC$_{16VOC}$**, **IVOC$_{30VOC}$** and **IVOC$_{65VOC}$** are the names of the different tested configurations. I suppose a part of the confusion also came from the use of the "SVOC/POA ratio" which was modified in the manuscript.

> *24. Page 13, lines 14-31: I was hoping this would be covered in the methods section.*

Several sensitivity tests were performed on the parameterization properties and IVOC emissions. For clarity, the choice was made to describe the model configuration used for the sensitivity tests in each of the sensitivity test sections. We still think that it improves the clarity of the manuscript and this was not changed in the revised version. A sentence was however added at the end of the VBS-GECKO parameterization description section (Section 2.2) to list all the sensitivity tests and send to the corresponding sensitivity test section. "Changes were then applied to this configuration to perform sensitivity tests of SOA formation on secondary organic compound

properties (solubility, reactivity with OH, NOx/HO$_2$ condition dependency and volatility) or SVOC and IVOC (S/IVOC) emissions. For a better readability, the details of these modifications are presented for each test before results in section 4 (properties) and section 5 (S/IVOC emissions)."

> *25. Figure 14: Could these also include the measurements?*

The Figure shows the temporal evolution of the total OA mass concentrations by sources. Measurements at the stations provide only OM in PM$_1$ (from ACSM), OC in PM$_{2.5}$ (from filter calcinations) or PM$_{2.5}$. The model/measurement comparison on the total OA mass is thus not possible.

> *26. Page 15, lines 15-21: Are newly discovered pathways for SOA from autooxidation reactions and acid-catalyzed pathways included? If not, could these be cited here to provide context?*

These pathways are not considered in GECKO-A, and so neither in the VBS-GECKO which is developed based on GECKO-A simulations. These recently identified processes can indeed influence SOA formation. These processes are currently not included in the VBS-GECKO parameterization, and could possibly explain the discrepancies between model and measurements. Sentences were added to the conclusion of the manuscript to discuss the possible origin of the model/measure discrepancies, and suggest some future development to improve the reliability of SOA parameterizations.

> *27. Page 17, line 6: I do not recollect where the volatility of the OA was discussed and how the authors conclude that LVOC and ELVOC make up most of the OA*

The volatility of OA is discussed based on Figure 10 in Section 4.4 and based on Figure S2 in the last paragraph of Section 6. The terms "LVOC" and "ELVOC" were indeed not defined in the text and were thus removed in the revised manuscript.

**References**

Ahmadov, R., McKeen, S. A., Robinson, A. L., Bahreini, R., Middlebrook, A. M., de Gouw, J. A., Meagher, J., Hsie, E.-Y., Edgerton, E., Shaw, S. and Trainer, M.: A volatility basis set model for summertime secondary

organic aerosols over the eastern United States in 2006, J. Geophys. Res. Atmos., 117(D6), n/a-n/a, doi:10.1029/2011jd016831, 2012.

Akherati, A., Cappa, C. D., Kleeman, M. J., Docherty, K. S., Jimenez, J. L., Griffith, S. M., Dusanter, S., Stevens, P. S. and Jathar, S. H.: Simulating secondary organic aerosol in a regional air quality model using the statistical oxidation model - Part 3: Assessing the influence of semi-volatile and intermediate-volatility organic compounds and NOx, Atmos. Chem. Phys., 19(7), 4561–4594, doi:10.5194/acp-19-4561-2019, 2019.

Bergström, R., Denier Van Der Gon, H. A. C., Prévôt, A. S. H., Yttri, K. E. and Simpson, D.: Modelling of organic aerosols over Europe (2002-2007) using a volatility basis set (VBS) framework: Application of different assumptions regarding the formation of secondary organic aerosol, Atmos. Chem. Phys., 12(18), 8499–8527, doi:10.5194/acp-12-8499-2012, 2012.

Bougiatioti, A., Nikolaou, P., Stavroulas, I, Kouvarakis, G., Weber, R., Nenes, A., Kanakidou, M. and Mihalopoulos, N.: Particle water and pH in the eastern Mediterranean: Source variability and implications for nutrient availability, Atmos. Chem. Phys., 16(7), 4579–4591, doi:10.5194/acp-16-4579-2016, 2016.

Boyd, A. A., Flaud, P. M., Daugey, N. and Lesclaux, R.: Rate constants for RO2 + HO2 reactions measured under a large excess of HO2, J. Phys. Chem. A, 107(6), 818–821, doi:10.1021/jp026581r, 2003.

Ciarelli, G., Aksoyoglu, S., El Haddad, I., Bruns, E. A., Crippa, M., Poulain, L., Äijälä, M., Carbone, S., Freney, E., O'Dowd, C., Baltensperger, U. and Prévôt, A. S. H.: Modelling winter organic aerosol at the European scale with CAMx: Evaluation and source apportionment with a VBS parameterization based on novel wood burning smog chamber experiments, Atmos. Chem. Phys., 17(12), 7653–7669, doi:10.5194/acp-17-7653-2017, 2017.

Couvidat, F. and Sartelet, K.: The Secondary Organic Aerosol Processor (SOAP v1.0) model: a unified model with different ranges of complexity based on the molecular surrogate approach, Geosci. Model Dev., 8(4), 1111–1138, doi:10.5194/gmd-8-1111-2015, 2015.

Couvidat, F., Debry, É., Sartelet, K. and Seigneur, C.: A hydrophilic/hydrophobic organic (H2O) aerosol model: Development, evaluation and sensitivity analysis, J. Geophys. Res., 117(D10), D10304, doi:10.1029/2011JD017214, 2012.

Couvidat, F., Bessagnet, B., Garcia-Vivanco, M., Real, E., Menut, L. and Colette, A.: Development of an inorganic and organic aerosol model (CHIMERE 2017β v1.0): seasonal and spatial evaluation over Europe, Geosci. Model Dev., 11(1), 165–194, doi:10.5194/gmd-11-165-2018, 2018.

Denier Van Der Gon, H. A. C., Bergström, R., Fountoukis, C., Johansson, C., Pandis, S. N., Simpson, D. and Visschedijk, A. J. H.: Particulate emissions from residential wood combustion in Europe - revised estimates and an evaluation, Atmos. Chem. Phys., 15(11), 6503–6519, doi:10.5194/acp-15-6503-2015, 2015.

Denjean, C., Formenti, P., Picquet-Varrault, B., Camredon, M., Pangui, E., Zapf, P., Katrib, Y., Giorio, C., Tapparo, A., Temime-Roussel, B., Monod, A., Aumont, B. and Doussin, J. F.: Aging of secondary organic aerosol generated from the ozonolysis of α-pinene: Effects of ozone, light and temperature, Atmos. Chem. Phys., 15(2), 883–897, doi:10.5194/acp-15-883-2015, 2015.

Guo, H., Xu, L., Bougiatioti, A., Cerully, K. M., Capps, S. L., Hite, J. R., Carlton, A. G., Lee, S. H., Bergin, M. H., Ng, N. L., Nenes, A. and Weber, R. J.: Fine-particle water and pH in the southeastern United States, Atmos. Chem. Phys., 15(9), 5211–5228, doi:10.5194/acp-15-5211-2015, 2015.

Hodzic, A., Madronich, S., Aumont, B., Lee-Taylor, J., Karl, T., Camredon, M. and Mouchel-Vallon, C.: Limited influence of dry deposition of semivolatile organic vapors on secondary organic aerosol formation in the urban plume, Geophys. Res. Lett., 40(12), 3302–3307, doi:10.1002/grl.50611, 2013.

Jenkin, M. E., Saunders, S. M. and Pilling, M. J.: The tropospheric degradation of volatile organic compounds: a protocol for mechanism development, Atmos. Environ., 31(1), 81–104, doi:10.1016/S1352-2310(96)00105-7, 1997.

La, Y. S., Camredon, M., Ziemann, P. J., Valorso, R., Matsunaga, A., Lannuque, V., Lee-Taylor, J., Hodzic, A., Madronich, S. and Aumont, B.: Impact of chamber wall loss of gaseous organic compounds on secondary organic aerosol formation: Explicit modeling of SOA formation from alkane and alkene oxidation, Atmos. Chem. Phys., 16(3), 1417–1431, doi:10.5194/acp-16-1417-2016, 2016.

Lane, T. E., Donahue, N. M. and Pandis, S. N.: Simulating secondary organic aerosol formation using the volatility basis-set approach in a chemical transport model, Atmos. Environ., 42(32), 7439–7451, doi:10.1016/j.atmosenv.2008.06.026, 2008.

Lannuque, V., Camredon, M., Couvidat, F., Hodzic, A., Valorso, R., Madronich, S., Bessagnet, B. and Aumont, B.: Exploration of the influence of environmental conditions on secondary organic aerosol formation and organic species properties using explicit simulations: development of the VBS-GECKO parameterization, Atmos. Chem. Phys., 18(18), 13411–13428, doi:10.5194/acp-18-13411-2018, 2018.

Lecœur, E. and Seigneur, C.: Dynamic evaluation of a multi-year model simulation of particulate matter concentrations over Europe, Atmos. Chem. Phys., 13(8), 4319–4337, doi:10.5194/acp-13-4319-2013, 2013.

Lee-Taylor, J., Madronich, S., Aumont, B., Baker, A., Camredon, M., Hodzic, A., Tyndall, G. S., Apel, E. and Zaveri, R. A.: Explicit modeling of organic chemistry and secondary organic aerosol partitioning for Mexico City and its outflow plume, Atmos. Chem. Phys., 11(24), 13219–13241, doi:10.5194/acp-11-13219-2011, 2011.

Lee-Taylor, J., Hodzic, A., Madronich, S., Aumont, B., Camredon, M. and Valorso, R.: Multiday production of condensing organic aerosol mass in urban and forest outflow, Atmos. Chem. Phys., 15(2), 595–615, doi:10.5194/acp-15-595-2015, 2015.

McVay, R. C., Zhang, X., Aumont, B., Valorso, R., Camredon, M., La, Y. S., Wennberg, P. O. and Seinfeld, J. H.: SOA formation from the photooxidation of α-pinene: Systematic exploration of the simulation of chamber data, Atmos. Chem. Phys., 16(5), 2785–2802, doi:10.5194/acp-16-2785-2016, 2016.

Mircea, M., Bessagnet, B., D'Isidoro, M., Pirovano, G., Aksoyoglu, S., Ciarelli, G., Tsyro, S., Manders, A., Bieser, J., Stern, R., Vivanco, M. G., Cuvelier, C., Aas, W., Prévôt, A. S. H., Aulinger, A., Briganti, G., Calori, G., Cappelletti, A., Colette, A., Couvidat, F., Fagerli, H., Finardi, S., Kranenburg, R., Rouïl, L., Silibello, C., Spindler, G., Poulain, L., Herrmann, H., Jimenez, J. L., Day, D. A., Tiitta, P. and Carbone, S.:

EURODELTA III exercise: An evaluation of air quality models' capacity to reproduce the carbonaceous aerosol, Atmos. Environ. X, 2(March 2018), 100018, doi:10.1016/j.aeaoa.2019.100018, 2019.

Murphy, B. N., Woody, M. C., Jimenez, J. L., Carlton, A. M. G., Hayes, P. L., Liu, S., Ng, N. L., Russell, L. M., Setyan, A., Xu, L., Young, J., Zaveri, R. A., Zhang, Q. and Pye, H. O. T.: Semivolatile POA and parameterized total combustion SOA in CMAQv5.2: Impacts on source strength and partitioning, Atmos. Chem. Phys., 17(18), 11107–11133, doi:10.5194/acp-17-11107-2017, 2017.

Robinson, A. L., Donahue, N. M., Shrivastava, M. K., Weitkamp, E. A., Sage, A. M., Grieshop, A. P., Lane, T. E., Pierce, J. R. and Pandis, S. N.: Rethinking Organic Aerosols: Semivolatile Emissions and Photochemical Aging, Science (80-. )., 315(5816), 1259–1262, doi:10.1126/science.1133061, 2007.

Valorso, R., Aumont, B., Camredon, M., Raventos-Duran, T., Mouchel-Vallon, C., Ng, N. L., Seinfeld, J. H., Lee-Taylor, J. and Madronich, S.: Explicit modelling of SOA formation from α-pinene photooxidation: Sensitivity to vapour pressure estimation, Atmos. Chem. Phys., 11(14), 6895–6910, doi:10.5194/acp-11-6895-2011, 2011.

Zhao, Y., Nguyen, N. T., Presto, A. A., Hennigan, C. J., May, A. A. and Robinson, A. L.: Intermediate Volatility Organic Compound Emissions from On-Road Diesel Vehicles: Chemical Composition, Emission Factors, and Estimated Secondary Organic Aerosol Production, Environ. Sci. Technol., 49(19), 11516–11526, doi:10.1021/acs.est.5b02841, 2015.

Zhao, Y., Nguyen, N. T., Presto, A. A., Hennigan, C. J., May, A. A. and Robinson, A. L.: Intermediate Volatility Organic Compound Emissions from On-Road Gasoline Vehicles and Small Off-Road Gasoline Engines, Environ. Sci. Technol., 50(8), 4554–4563, doi:10.1021/acs.est.5b06247, 2016.

---

## Author Response (AR2)

**Author Responses to Editor and comments on**

**Modelling organic aerosol over Europe in summer conditions with the VBS-GECKO parameterization: sensitivity to secondary organic compound properties and IVOC emissions**

Victor Lannuque[1,2,3,a], Florian Couvidat[2], Marie Camredon[1], Bernard Aumont[1] and Bertrand Bessagnet[2,b]

[1] LISA, UMR CNRS 7583, IPSL, Université Paris Est Créteil and Université Paris Diderot, 94010 Créteil Cedex, France.
[2] INERIS, National Institute for Industrial Environment and Risks, Parc Technologique ALATA, 60550 Verneuil-en-Halatte, France.
[3] Agence de l'Environnement et de la Maîtrise de l'Energie, 20 avenue du Grésillé - BP 90406, 49004 Angers Cedex 01, France.
[a] Now at: Laboratoire d'Aérologie, Université de Toulouse, CNRS, UPS, Toulouse, France.
[b] Now at: Hangzhou Futuris Environmental Technology Co. Ltd, Zhejiang Overseas High-Level Talent Innovation Park, No. 998 WenYi Road, 311121, Hangzhou, Zhejiang, China.

*Correspondence to*: Victor Lannuque (victor.lannuque@gmail.com) and Florian Couvidat (florian.couvidat@ineris.fr)

We thank the editor for his last comments on the manuscript.

**Reponses to Editor**

To take into account the remarks of the referee relayed by the editor regarding the parameterization and the state of knowledge in SOA formation, we added two paragraphs to the introduction to put into perspective the work carried out, and present some limits of the approach used (Changes are highlighted in yellow in the last version of the manuscript which follows the responses).

Concerning the description of the parameterization in section 2.2.2, the authors prefer to leave the section as it is, developing it more would come close to the previous paper describing in detail the development of the parameterization (Lannuque et al., 2018) which is not the subject of this article; synthesizing it would force us to omit information and bring us back to the very first version of the manuscript, before the referees' request to detail the section.

[revised manuscript text omitted]

The development of the VBS-GECKO parameterization explores another track using the results of an explicit model to represent the organic gas-phase chemistry and gas/particles mass transfer, instead of atmospheric chamber data. The VBS-GECKO parameterization for SOA formation (Lannuque et al., 2018) is a VBS-type parameterization with gaseous aging.

15   VBS-GECKO was optimized based on box modeling results using explicit oxidation mechanisms generated with the Generator for Explicit Chemistry and Kinetics of Organics in the Atmosphere (GECKO-A) modeling tool (Aumont et al, 2005; Camredon et al, 2007). Lannuque et al. (2018) have shown that the VBS-GECKO parameterization is coherent compared to the explicit GECKO-A chemical mechanism. The reliability of the VBS-GECKO parameterization is by design directly linked to the accuracy of the GECKO-A mechanisms. The accuracy of the GECKO-A mechanisms to represent SOA

20   formation has been evaluated against around 50 chamber experiments (Denjean et al, 2015; La et al, 2016 ; McVay et al, 2016; Valorso et al, 2011). Some processes relevant for SOA formation in the atmosphere can however be misrepresented in the GECKO-A mechanisms, or are just not included (such as gaseous autoxidation reactions or the reactivity in the condensed phase). The reliability of the VBS-GECKO parameterization to represent SOA formation observed in the atmosphere has thus to be evaluated.

[revised manuscript text omitted]

Ehn, M., Thornton, J. A., Kleist, E., Sipila, M., Junninen, H., Pullinen, I., Springer, M., Rubach, F., Tillmann, R., Lee, B., Lopez-Hilfiker, F., Andres, S., Acir, I. H., Rissanen, M., Jokinen, T., Schobesberger, S., Kangasluoma, J., Kontkanen, J., Nieminen, T., Kurten, T., Nielsen, L. B., Jorgensen, S., Kjaergaard, H. G., Canagaratna, M., Dal Maso, M., Berndt, T., Petaja, T., Wahner, A., Kerminen, V. M., Kulmala, M., Worsnop, D. R., Wildt, J. and Mentel, T. F.: A large source of low-volatility secondary organic aerosol, Nature, 506(7489), 476–479, https://doi.org/10.1038/Nature13032, 2014.

[revised manuscript text omitted]